# The vaginal immunoproteome for the prediction of spontaneous preterm birth: A retrospective longitudinal study

Zachary Shaffer[1,2,3], Roberto Romero[1,4,5]*, Adi L Tarca[1,2,6,7]*, Jose Galaz[1,2,8], Marcia Arenas-Hernandez[1,2], Dereje W Gudicha[1,2], Tinnakorn Chaiworapongsa[1,2], Eunjung Jung[1,2‡], Manaphat Suksai[1,2], Kevin R Theis[1,2,9]*, Nardhy Gomez-Lopez[1,2,7,9]*#§

[1]Pregnancy Research Branch, Division of Obstetrics and Maternal-Fetal Medicine, Division of Intramural Research, Eunice Kennedy Shriver National Institute of Child Health and Human Development, National Institutes of Health, US Department of Health and Human Services (NICHD/NIH/DHHS), Bethesda, United States; [2]Department of Obstetrics and Gynecology, Wayne State University School of Medicine, Detroit, United States; [3]Department of Physiology, Wayne State University School of Medicine, Detroit, United States; [4]Department of Obstetrics and Gynecology, University of Michigan, Ann Arbor, United States; [5]Department of Epidemiology and Biostatistics, Michigan State University, East Lansing, United States; [6]Department of Computer Science, Wayne State University College of Engineering, Detroit, United States; [7]Center for Molecular Medicine and Genetics, Wayne State University, Detroit, United States; [8]Division of Obstetrics and Gynecology, Faculty of Medicine, Pontificia Universidad Católica de Chile, Santiago, Chile; [9]Department of Biochemistry, Microbiology and Immunology, Wayne State University School of Medicine, Detroit, United States

*For correspondence: romeror@mail.nih.gov (RR); atarca@med.wayne.edu (ALT); ktheis@med.wayne.edu (KRT); nardhy@wustl.edu (NG-L)

Present address: ‡Department of Obstetrics and Gynecology, Busan Paik Hospital, Inje University College of Medicine, Busan, South Korea; §Center for Reproductive Health Sciences, Department of Obstetrics and Gynecology & Pathology and Immunology, Washington University School of Medicine, St. Louis, United States

#Lead contact

Competing interest: The authors declare that no competing interests exist.

## Abstract

**Background:** Preterm birth is the leading cause of neonatal morbidity and mortality worldwide. Most cases of preterm birth occur spontaneously and result from preterm labor with intact (spontaneous preterm labor [sPTL]) or ruptured (preterm prelabor rupture of membranes [PPROM]) membranes. The prediction of spontaneous preterm birth (sPTB) remains underpowered due to its syndromic nature and the dearth of independent analyses of the vaginal host immune response. Thus, we conducted the largest longitudinal investigation targeting vaginal immune mediators, referred to herein as the immunoproteome, in a population at high risk for sPTB.

**Methods:** Vaginal swabs were collected across gestation from pregnant women who ultimately underwent term birth, sPTL, or PPROM. Cytokines, chemokines, growth factors, and antimicrobial peptides in the samples were quantified via specific and sensitive immunoassays. Predictive models were constructed from immune mediator concentrations.

**Results:** Throughout uncomplicated gestation, the vaginal immunoproteome harbors a cytokine network with a homeostatic profile. Yet, the vaginal immunoproteome is skewed toward a pro-inflammatory state in pregnant women who ultimately experience sPTL and PPROM. Such an inflammatory profile includes increased monocyte chemoattractants, cytokines indicative of macrophage and T-cell activation, and reduced antimicrobial proteins/peptides. The vaginal immunoproteome has improved predictive value over maternal characteristics alone for identifying women at risk for early (<34 weeks) sPTB.

**Conclusions:** The vaginal immunoproteome undergoes homeostatic changes throughout gestation and deviations from this shift are associated with sPTB. Furthermore, the vaginal immunoproteome can be leveraged as a potential biomarker for early sPTB, a subset of sPTB associated with extremely adverse neonatal outcomes.

**Funding:** This research was conducted by the Perinatology Research Branch, Division of Obstetrics and Maternal-Fetal Medicine, Division of Intramural Research, *Eunice Kennedy Shriver* National Institute of Child Health and Human Development, National Institutes of Health, U.S. Department of Health and Human Services (NICHD/NIH/DHHS) under contract HHSN275201300006C. ALT, KRT, and NGL were supported by the Wayne State University Perinatal Initiative in Maternal, Perinatal and Child Health.

## Editor's evaluation

This study presents important findings about the vaginal immunoproteome throughout pregnancy in cases of term or preterm birth, showing dynamic differences over time and between birth groups. The findings are solid and convincing in demonstrating the clinical usefulness as informative biomarker material to monitor pregnancy health and risk for preterm birth

## Introduction

Preterm birth, defined as live birth before 37 weeks of gestation (*American College of Obstetricians and Gynecologists, 2021*), afflicts 1 of every 10 children born worldwide (*Chawanpaiboon et al., 2019*; *Martin and Hamilton, 2021*). The economic burden of preterm birth amounts to more than $25.2 billion in healthcare costs annually in the United States alone (*Waitzman et al., 2021*). Two-thirds of preterm births occur spontaneously (i.e., spontaneous preterm birth [sPTB]) (*Goldenberg et al., 2008*), while the remainder are due to evident clinical conditions (e.g., preeclampsia and intra-uterine growth restriction) that jeopardize maternal–fetal well-being and require medically indicated delivery (i.e., iatrogenic preterm birth) (*Goldenberg et al., 2008*; *Romero et al., 2014a*). sPTB can be further subdivided into spontaneous preterm labor with intact membranes (sPTL) and preterm prelabor rupture of membranes (PPROM) (*Goldenberg et al., 2008*). The pathogenesis of sPTL and PPROM can include local inflammatory processes (*Romero et al., 2014a*); yet, each is considered a syndrome with distinct underlying mechanisms of disease (*Goldenberg et al., 2008*; *Romero et al., 2014a*) and thus requiring different clinical managements (*American College of Obstetricians and Gynecologists, 2016*; *Dvorakova and Ivankova, 2020*). Therefore, research focused on the prediction and prevention of sPTB should account for the distinct inflammatory nature of sPTL and PPROM.

Multiple attempts have been made to predict sPTB using data from noninvasive sampling coupled with omics platforms such as genomics (*Frey et al., 2016*; *Modi et al., 2017*; *Modi et al., 2018*; *Jain et al., 2022*), transcriptomics (*Ngo et al., 2018*; *Jehan et al., 2020*; *Peterson et al., 2020*; *Tarca et al., 2021*; *Camunas-Soler et al., 2022*), and proteomics (*Peterson et al., 2020*; *Jehan et al., 2020*; *Tarca et al., 2021*; *Tiensuu et al., 2022*). Yet, to date, assessment of cervical length remains the strongest and most cost-effective predictor of sPTB: women with a sonographic short cervix (≤ 25 mm) are sixfold more likely to deliver a preterm neonate (*Iams et al., 1996*), and tailored treatment with natural progesterone reduces such risk by half (*Fonseca et al., 2007*; *Hassan et al., 2011*; *Romero et al., 2014b*; *Romero et al., 2016*; *Romero et al., 2017*; *Conde-Agudelo et al., 2018*; *Romero et al., 2018*). Indeed, personalized cervical length assessment that accounts for maternal characteristics and obstetrical history was shown to improve prediction relative to raw cervical length data (*Gudicha et al., 2021*); however, additional biomarkers are still needed to further increase prediction performance. Growing evidence has fostered the hypothesis that cervical disease is associated with changes in the vaginal ecosystem (*Kindinger et al., 2017*; *Witkin et al., 2019*; *Di Paola et al., 2020*; *Gerson et al., 2020*). Thus, intensive investigation has focused on the vaginal microbiome and its potential utility for predicting sPTB (*Donders et al., 2009*; *DiGiulio et al., 2015*; *Kindinger et al., 2017*; *Freitas et al., 2018*; *Fettweis et al., 2019*; *Odogwu et al., 2021*; *Payne et al., 2021*; *Flaviani et al., 2021*; *Pruski et al., 2021*; *Kumar et al., 2021*). However, models utilizing vaginal microbiome data alone have displayed weak predictive power (*Freitas et al., 2018*; *Kumar et al., 2021*; *Pruski et al., 2021*), potentially due to sample size, sequencing depth, and ethnicity-driven differences in

**eLife digest** Human pregnancies last 40 weeks on average. Preterm births, defined as live births before 37 weeks, occur in about one in ten pregnancies. Being born too early is the main cause of a number of diseases and death in newborn babies.

Preterm births are further divided into those that happen early – before 34 weeks – and those that happen late – between 34 and 37 weeks. There are also differences between preterm births in which the amniotic sac ruptures before or after the start of labor.

Although several factors can lead to spontaneous preterm birth, bacteria getting into the amniotic fluid around the fetus are a well-known trigger. These bacteria usually come from the vagina. In the past, researchers have studied the number and types of bacteria in the vagina of people who had a normal pregnancy and those that had a preterm birth to predict who is more at risk of preterm birth.

However, predictions based only on data about bacteria have been less useful so far. Instead, it might be better to investigate a person's immune response during pregnancy. Shaffer et al. addressed this gap by asking whether measuring the levels of proteins involved in the immune response could help predict preterm births.

Shaffer et al. collected vaginal fluids from 739 individuals of predominately African American ethnicity with an average BMI of 28.7 – representing a population at high risk for spontaneous preterm birth. The swabs were taken at multiple points during their pregnancy, and 31 different immune-related proteins in those fluids were measured. The researchers further noted whether these individuals had a normal or a preterm birth.

The data showed that, compared to normal births, preterm births are associated with higher levels of proteins that attract white blood cells and promote inflammation, such as IL-6 and IL-1β. Vaginal fluids from individuals who went on to have an early preterm birth where the amniotic sac ruptured before labor, contained lower levels of proteins known as defensins, which defend the body from bacteria.

With these new data from vaginal swabs, Shaffer et al. could make better predictions about the likelihood of preterm birth in general and early preterm birth with the amniotic sac ruptured before labor. For the latter scenario, the predictions were not improved when combining immune protein data with other characteristics of the pregnant person, such as age.

These findings suggest that clinicians may be able to use measurements of immune-related proteins to help predict preterm births, so that pregnant individuals at high risk can receive extra care. Further research will have to validate the data and determine whether the findings apply more widely.

microbial community profile. Recent models, however, have leveraged vaginal host–microbe interactions by incorporating the determination of immune mediators, which improved the prediction of sPTB (*Elovitz et al., 2019*; *Fettweis et al., 2019*; *Pruski et al., 2021*; *Kumar et al., 2021*). Nonetheless, an extensive interrogation of the vaginal soluble immune response (i.e., the vaginal immunoproteome), with consideration of the distinction between sPTL and PPROM cases, has not been undertaken.

Herein, we conducted the largest assessment, based on the study's scale, longitudinal nature, and depth of immunological mediators evaluated, of the vaginal immunoproteome during uncomplicated and complicated pregnancies. Importantly, our determinations were performed in vaginal samples collected during all three trimesters and considered sPTB, including its two subsets (i.e., sPTL and PPROM), as well as the timing of delivery (i.e., early and late sPTB). Furthermore, the immunological mediators evaluated were selected for their relevance to key biological processes in the vaginal ecosystem. Additionally, we used machine learning approaches and cross-validation to generate and assess predictive models for sPTL and PPROM in our high-risk population.

## Results

### Characteristics of the study population

This study represents the largest (N = 739 women, n = 2819 samples) and most comprehensive (31 immune mediators) profiling of the soluble immune response in vaginal fluid throughout well-characterized pregnancies that culminated in the delivery of a term neonate (controls) or those that

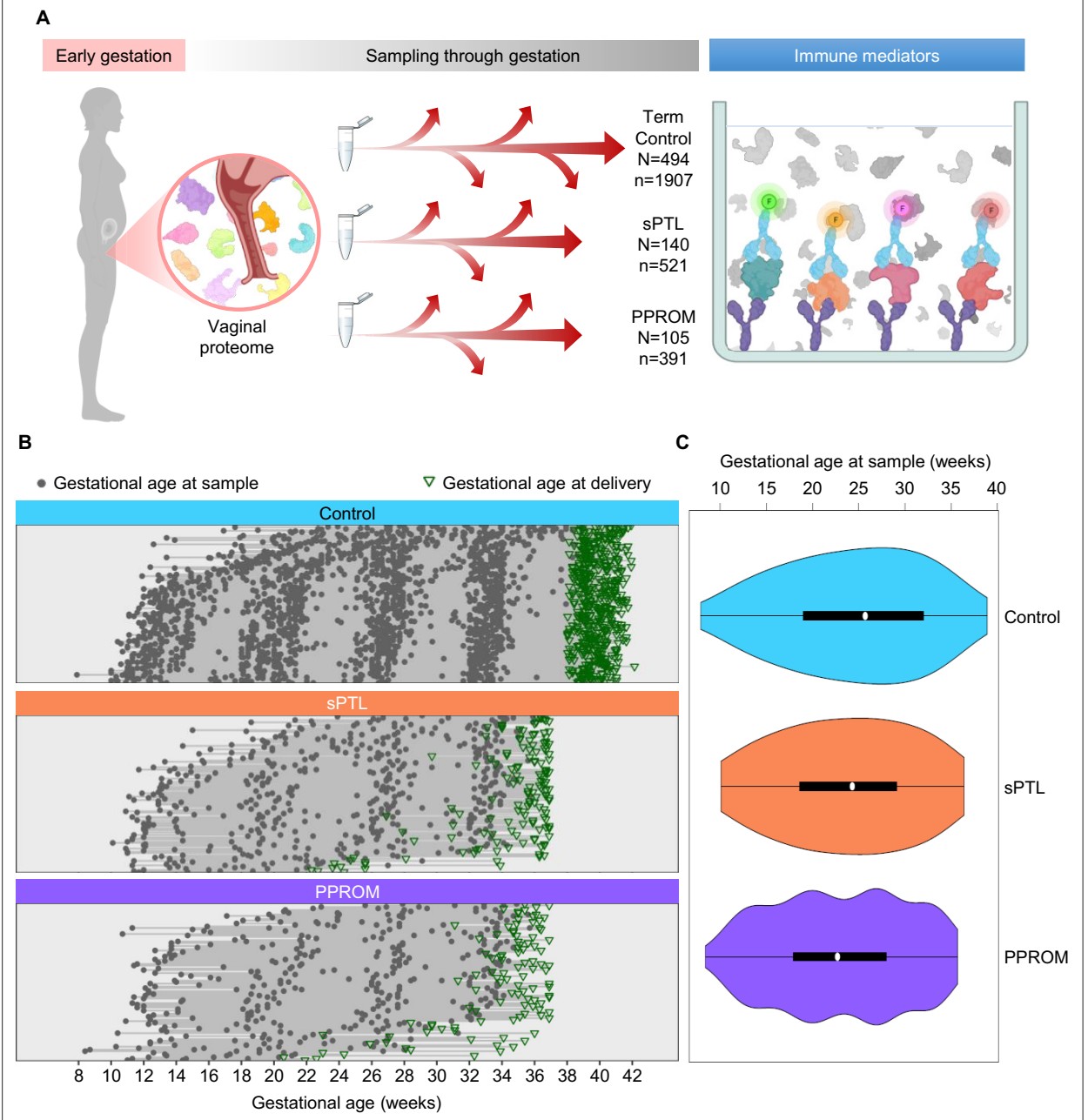

**Figure 1.** Longitudinal vaginal sampling of women with a term or preterm delivery. (**A**) Representative diagram showing the collection of vaginal swabs throughout gestation from women who underwent uncomplicated term birth (control; N = 494 subjects, n = 1907 swabs), spontaneous preterm labor with intact membranes (sPTL; N = 140, n = 521), or preterm prelabor rupture of membranes (PPROM; N = 105, n = 391) to quantify the concentrations of cytokines, chemokines, and other immune mediators in the cervico-vaginal fluid by immunoassay. (**B**) Plots showing the distribution of swabs collected throughout gestation (weeks) for each study group. Gray dots represent gestational ages when vaginal swabs were taken, and green triangles show the gestational age at delivery. (**C**) Violin plots showing sample frequency across gestational age (weeks) for each study group.

resulted in sPTB (sPTL and PPROM cases) (*Figure 1A*). Specifically, we enrolled pregnant women from whom vaginal swabs were collected at different points throughout gestation. Patients were followed until delivery, after which we selected women who underwent sPTL or PPROM with delivery before 37 weeks of gestation (preterm birth) and women with a normal term delivery as controls (matched by gestational age at sampling at a 2:1 ratio). All patients had a singleton pregnancy without fetal

**Table 1.** Demographics and clinical characteristics of the study groups.

Demographics and clinical information are presented as a median (with interquartile ranges) or as proportions (n/N). Differences between study groups and the control group were assessed by the Mann–Whitney $U$ test for continuous data or the Fisher's exact test for categorical data.

| | Term control (n = 494) | Spontaneous preterm birth (n = 245) | | | |
| | | sPTL (n = 140) | p | PPROM (n = 105) | p |
|---|---|---|---|---|---|
| Maternal age (years) | 24 (21–27) | 23.5 (21-27) | 0.37 | 25 (22–30) | 0.02 |
| BMI (kg/m$^2$) | 27.5 (22.7–33.8) | 26.4 (22.5–31.7) | 0.05 | 28.2 (23.2–33.3) | 0.79 |
| Maternal race | | | 0.09 | | 0.4 |
| African American | 94.5% (467/494) | 90% (126/140) | | 98.1% (103/105) | |
| White | 1.8% (9/494) | 5% (7/140) | | 1% (1/105) | |
| Other | 3.7 (18/494) | 5% (7/140) | | 1% (1/105) | |
| Nulliparity | 20.4% (101/494) | 12.9% (18/140) | 0.02 | 19% (20/105) | 0.79 |
| History of preterm birth | 11.1% (55/494) | 39.3% (55/140) | <0.001 | 32.4% (34/105) | <0.001 |
| Gestational age at delivery (weeks) | 39.6 (39–40.4) | 35.7 (33.7–36.6) | <0.001 | 35 (32.6–35.9) | <0.001 |
| Birthweight (g) | 3300 (3090–3593) | 2363 (1995–2748) | <0.001 | 2235 (1760–2575) | <0.001 |
| Maternal Inflammatory response | 0% (0/494) | 21.1% (28/133)* | <0.001 | 22.4% (22/98)* | <0.001 |
| Fetal inflammatory response | 4.3% (21/494) | 20.3% (27/133)* | <0.001 | 28.5% (28/98)* | <0.001 |
| Early sPTB (<34 weeks) | – | 27.1% (38/140) | | 36.2% (38/105) | |
| Late sPTB (34–36$^{+6}$) weeks | – | 72.9% (102/140) | | 63.8% (67/105) | |

BMI = body mass index; sPTL = spontaneous preterm labor; sPTB = spontaneous preterm birth; PPROM = preterm prelabor rupture of membranes.
*Seven missing data.

anomalies and had at least three vaginal samples available. Patients were classified into three study groups according to pregnancy outcome and diagnosis: (i) women who delivered at term (494 patients, 1907 samples); (ii) women who underwent sPTL (140 patients, 521 samples); and (iii) women who experienced PPROM (105 patients, 391 samples) (*Figure 1B and C*). By design, gestational age at sampling was similar among the study groups (*Figure 1C*). The clinical characteristics of the study population are shown in *Table 1*. As expected (*Goldenberg et al., 2008*), the prevalence of sPTL or PPROM was higher among women with a history of sPTB. Of note, this study addressed a high-risk population primarily composed of African-American women with an average body mass index (BMI) of 28.7 kg/m$^2$. Hence, the current study represents the largest survey of the vaginal immunoproteome in a population at high risk for sPTB.

## The vaginal immunoproteome is tightly regulated during normal gestation

First, by profiling 31 immune mediators, including two antimicrobial molecules, we established the gestational age-dependent changes in the vaginal immunoproteome during normal gestation. Given that the immunobiology of preterm birth is syndromic in nature (*Romero et al., 2014a*; *Gomez-Lopez et al., 2022a*), in this study we classified the immune mediators measured in the vaginal fluid into six categories based on their established role and/or their potential cellular source: pro-inflammatory cytokines (IL-6, IL-1β, IL-16, CXCL8, TNF, IFN-γ, IL-1α, and LT-α), chemoattractants of monocytes (herein referred to as monocyte chemokines; CCL2, CCL3, and CCL4), macrophage cytokines (IL-12/IL-23p40, IL-12p70, and IL-15), T-cell cytokines and chemokines (IL-2, IL-4, IL-17A, IL-10, IL-13, IL-5, CXCL10, and CCL17), antimicrobial peptides (AMPs; β-defensin-2 and secretory leukocyte peptidase inhibitor [SLPI]), and growth factors (VEGF and GM-CSF). All immune mediators, except for CCL11, CCL13, CCL22, and CCL26, were present at detectable levels in the majority of our study samples (*Table 2*) and were included in our analyses. Importantly, one-third of these mediators showed an association with gestational age (*Figure 2*; *Figure 2—source data 1 and 2*), whereas the remaining were

**Table 2.** Immune mediator assay sensitivity and proportion outside of range of detection.

Vaginal immune mediator proportions below the limit of detection and greater than 2 * 99th percentile. Chemokines listed in red were excluded from analyses due to a majority of samples being below the limit of detection.

| Immune mediator | Sensitivity of detection | Control | | | | sPTL | | | | PPROM | | | |
|---|---|---|---|---|---|---|---|---|---|---|---|---|---|
| | | Below limit | Below limit % | Above limit | Above limit % | Below limit | Below limit % | Above limit | Above limit % | Below limit | Below limit % | Above limit | Above limit % |
| IFN-γ | 0.368 pg/mL | 515.00 | 27.01 | 7.00 | 0.37 | 103.00 | 19.77 | 1.00 | 0.19 | 83.00 | 21.23 | 1.00 | 0.26 |
| IL-1β | 0.152 pg/mL | 0.00 | 0.00 | 1.00 | 0.05 | 4.00 | 0.77 | 1.00 | 0.19 | 1.00 | 0.26 | 0.00 | 0.00 |
| IL-2 | 0.219 pg/mL | 29.00 | 1.52 | 2.00 | 0.10 | 10.00 | 1.92 | 1.00 | 0.19 | 6.00 | 1.53 | 0.00 | 0.00 |
| IL-4 | 0.028 pg/mL | 147.00 | 7.71 | 2.00 | 0.10 | 30.00 | 5.76 | 2.00 | 0.38 | 30.00 | 7.67 | 2.00 | 0.51 |
| IL-6 | 0.113 pg/mL | 43.00 | 2.25 | 0.00 | 0.00 | 10.00 | 1.92 | 2.00 | 0.38 | 8.00 | 2.05 | 3.00 | 0.77 |
| CXCL8 | 0.101 pg/mL | 0.00 | 0.00 | 0.00 | 0.00 | 4.00 | 0.77 | 1.00 | 0.19 | 0.00 | 0.00 | 0.00 | 0.00 |
| IL-10 | 0.054 pg/mL | 34.00 | 1.78 | 2.00 | 0.10 | 10.00 | 1.92 | 3.00 | 0.58 | 5.00 | 1.28 | 2.00 | 0.51 |
| IL-12p70 | 0.084 pg/mL | 178.00 | 9.33 | 0.00 | 0.00 | 35.00 | 6.72 | 1.00 | 0.19 | 42.00 | 10.74 | 0.00 | 0.00 |
| IL-13 | 0.119 pg/mL | 55.00 | 2.88 | 0.00 | 0.00 | 9.00 | 1.73 | 2.00 | 0.38 | 16.00 | 4.09 | 0.00 | 0.00 |
| TNF | 0.108 pg/mL | 94.00 | 4.93 | 2.00 | 0.10 | 14.00 | 2.69 | 4.00 | 0.77 | 23.00 | 5.88 | 2.00 | 0.51 |
| GM-CSF | 0.200 pg/mL | 176.00 | 9.23 | 4.00 | 0.21 | 46.00 | 8.83 | 2.00 | 0.38 | 30.00 | 7.67 | 2.00 | 0.51 |
| IL-1α | 0.089 pg/mL | 15.00 | 0.79 | 8.00 | 0.42 | 14.00 | 2.69 | 1.00 | 0.19 | 5.00 | 1.28 | 3.00 | 0.77 |
| IL-5 | 0.162 pg/mL | 324.00 | 16.99 | 4.00 | 0.21 | 102.00 | 19.58 | 3.00 | 0.58 | 82.00 | 20.97 | 1.00 | 0.26 |
| IL-7 | 0.124 pg/mL | 772.00 | 40.48 | 3.00 | 0.16 | 190.00 | 36.47 | 4.00 | 0.77 | 141.00 | 36.06 | 1.00 | 0.26 |
| IL-12/IL-23p40 | 0.408 pg/mL | 211.00 | 11.06 | 8.00 | 0.42 | 48.00 | 9.21 | 1.00 | 0.19 | 34.00 | 8.70 | 2.00 | 0.51 |
| IL-15 | 0.111 pg/mL | 707.00 | 37.07 | 4.00 | 0.21 | 159.00 | 30.52 | 1.00 | 0.19 | 142.00 | 36.32 | 6.00 | 1.53 |
| IL-16 | 0.478 pg/mL | 123.00 | 6.45 | 4.00 | 0.21 | 31.00 | 5.95 | 1.00 | 0.19 | 26.00 | 6.65 | 1.00 | 0.26 |
| IL-17A | 0.381 pg/mL | 247.00 | 12.95 | 10.00 | 0.52 | 42.00 | 8.06 | 4.00 | 0.77 | 46.00 | 11.76 | 4.00 | 1.02 |
| LT-α | 0.095 pg/mL | 957.00 | 50.18 | 4.00 | 0.21 | 236.00 | 45.30 | 3.00 | 0.58 | 201.00 | 51.41 | 0.00 | 0.00 |
| VEGF | 0.264 pg/mL | 7.00 | 0.37 | 2.00 | 0.10 | 9.00 | 1.73 | 2.00 | 0.38 | 12.00 | 3.07 | 1.00 | 0.26 |
| CCL11/Eotaxin | 0.480 pg/mL | 1504.00 | 78.87 | 0.00 | 0.00 | 379.00 | 72.74 | 0.00 | 0.00 | 294.00 | 75.19 | 1.00 | 0.26 |
| CCL26/Eotaxin-3 | 1.099 pg/mL | 1480.00 | 77.61 | 7.00 | 0.37 | 394.00 | 75.62 | 3.00 | 0.58 | 309.00 | 79.03 | 0.00 | 0.00 |
| CXCL8 (HA) | 22.575 pg/mL | 84.00 | 4.40 | 0.00 | 0.00 | 11.00 | 2.11 | 1.00 | 0.19 | 23.00 | 5.88 | 0.00 | 0.00 |
| CXCL10 | 0.119 pg/mL | 4.00 | 0.21 | 2.00 | 0.10 | 2.00 | 0.38 | 2.00 | 0.38 | 0.00 | 0.00 | 0.00 | 0.00 |
| CCL2 | 0.072 pg/mL | 10.00 | 0.52 | 2.00 | 0.10 | 4.00 | 0.77 | 2.00 | 0.38 | 0.00 | 0.00 | 4.00 | 1.02 |
| CCL13/MCP-4 | 0.230 pg/mL | 1693.00 | 88.78 | 2.00 | 0.10 | 438.00 | 84.07 | 7.00 | 1.34 | 332.00 | 84.91 | 5.00 | 1.28 |
| CCL22/MDC | 1.740 pg/mL | 985.00 | 51.65 | 3.00 | 0.16 | 245.00 | 47.02 | 2.00 | 0.38 | 197.00 | 50.38 | 0.00 | 0.00 |
| CCL3 | 0.319 pg/mL | 718.00 | 37.65 | 2.00 | 0.10 | 134.00 | 25.72 | 2.00 | 0.38 | 116.00 | 29.67 | 2.00 | 0.51 |
| CCL4 | 0.226 pg/mL | 221.00 | 11.59 | 2.00 | 0.10 | 33.00 | 6.33 | 2.00 | 0.38 | 32.00 | 8.18 | 2.00 | 0.51 |
| CCL17 | 0.375 pg/mL | 192.00 | 10.07 | 0.00 | 0.00 | 45.00 | 8.64 | 2.00 | 0.38 | 38.00 | 9.72 | 0.00 | 0.00 |
| SLPI | 20.336 pg/mL | 0.00 | 0.00 | 0.00 | 0.00 | 0.00 | 0.00 | 0.00 | 0.00 | 0.00 | 0.00 | 0.00 | 0.00 |
| β-defensin-2 | 0.007 ng/mL | 0.00 | 0.00 | 0.00 | 0.00 | 0.00 | 0.00 | 0.00 | 0.00 | 0.00 | 0.00 | 0.00 | 0.00 |

sPTL = spontaneous preterm labor; PPROM = preterm prelabor rupture of membranes; HA = high affinity.

unaffected (*Figure 2—figure supplements 1 and 2 , Figure 2—source data 1 and 2* ). Specifically, the vaginal concentrations of IL-1β (*Figure 2*), IL-1α (*Figure 2B*), CXCL8 (*Figure 2C*), IL-2 (*Figure 2D*), IL-12/IL-23p40 (*Figure 2E*), GM-CSF (*Figure 2F*), and β-defensin-2 (*Figure 2G*) declined as gestation progressed. On the other hand, CCL17 (*Figure 2H*), CXCL10 (*Figure 2I*), and VEGF (*Figure 2J*) increased with gestational age. A tendency for increased concentrations of SLPI with advancing

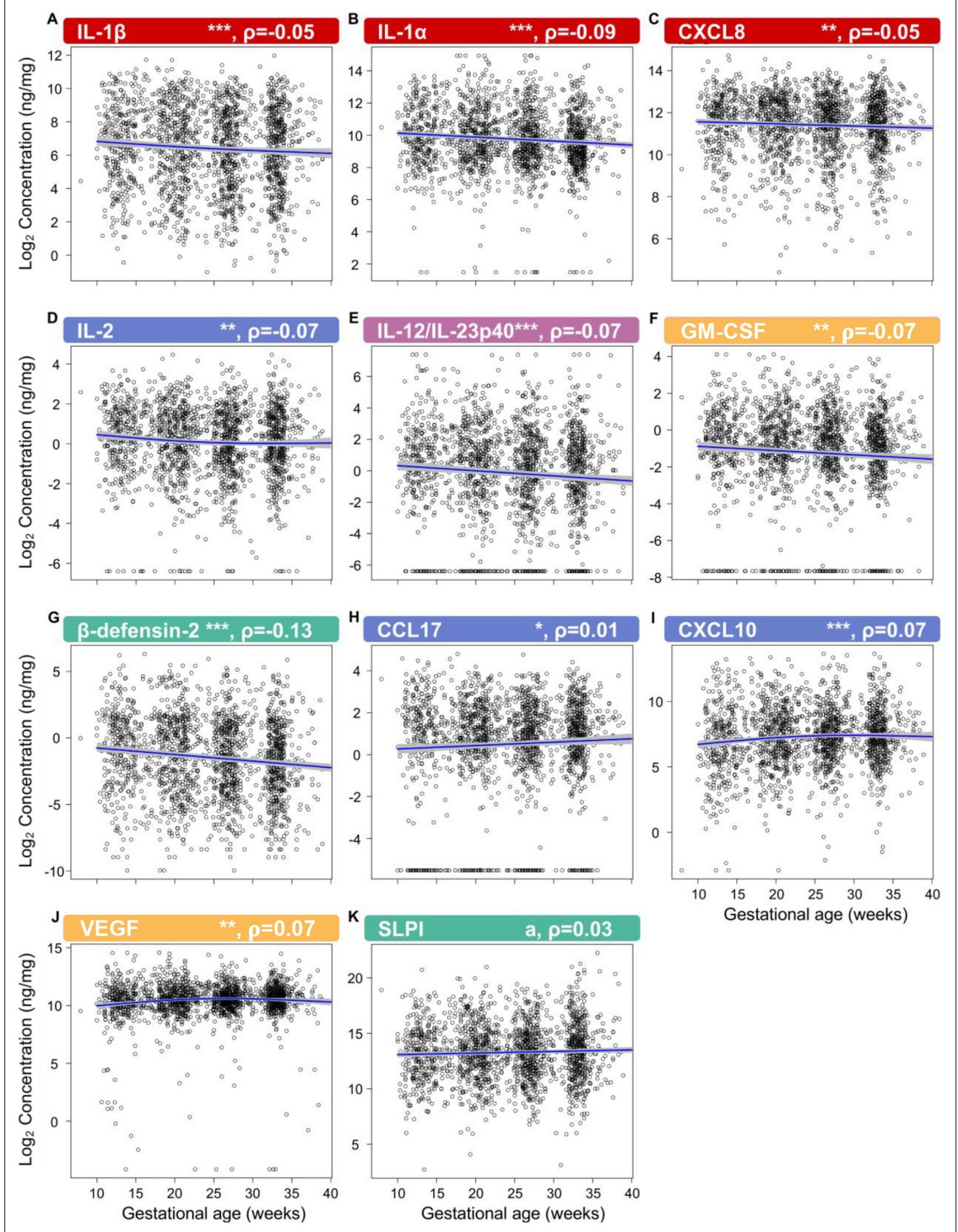

**Figure 2.** The vaginal immunoproteome is finely regulated during normal gestation. Linear mixed effects modeling was used to determine changes in (**A**) IL-1β, (**B**) IL-1α, (**C**) CXCL8, (**D**) IL-2, (**E**) IL-12/IL-23p40, (**F**) GM-CSF, (**G**) β-defensin-2, (**H**) CCL17, (**I**) CXCL10, (**J**) VEGF, and (**K**) SLPI throughout pregnancies, resulting in term birth. Immune mediator labels denote category: pro-inflammatory cytokines (red), T-cell cytokine response (blue), macrophage cytokine response (purple), chemokines (green), growth factors (orange), and antimicrobial peptides (AMPs) (teal). Each black circle represents the protein concentration of one sample. Blue lines represent the linear fit, with gray shaded areas representing the 95% confidence interval of the mean. Scatterplot trends were estimated using generalized additive models with spline transformations of gestational age. *p<0.05 and q<0.1, **p<0.01 and q<0.1, ***p<0.001 and q<0.1, ªp<0.05 and q>0.1. Correlation coefficients are shown above each figure as rho-values ( $\rho$ ).

The online version of this article includes the following source data and figure supplement(s) for figure 2:

*Figure 2 continued on next page*

*Figure 2 continued*

**Source data 1.** Vaginal immune mediator shifts across gestational age in term uncomplicated pregnancies.

**Source data 2.** The 10th, 50th, and 95th quantiles of vaginal immune mediator concentrations weekly across gestation, resulting in term birth.

**Figure supplement 1.** Vaginal pro-inflammatory mediators, monocyte chemoattractants, and macrophage cytokines unaltered with gestational age in women who ultimately delivered at term.

**Figure supplement 2.** Vaginal T-cell cytokines and the growth factor IL-7 that are unaltered with gestational age in women who ultimately delivered at term.

gestational age was also observed (*Figure 2K*). These data indicate that the vaginal immunoproteome undergoes specific changes during normal pregnancy, which involve a decline in pro-inflammatory mediators, an increase in T cell-associated chemokines, and the tight regulation of growth factors and defensins. Together, these findings suggest that the vaginal cytokine network in normal pregnancy is modestly modulated during gestation, which prompted us to investigate whether a deviation from these changes occurs in women who ultimately undergo sPTB.

## The vaginal immunoproteome of women with sPTB has a pro-inflammatory profile

To interrogate whether alterations in the vaginal immunoproteome are observed in women destined to undergo sPTB, immune mediator data were quantified in vaginal swabs collected longitudinally from early to late gestation (in the interval of 8–37 weeks) from women who ultimately delivered term or preterm. Linear mixed effects modeling controlling for gestational age at sampling, maternal age, BMI, parity, and history of preterm birth was used to compare the concentrations of immune mediators between cases and controls (*Figure 3*, *Figure 4*, *Figure 5*, *Figure 6*, *Figure 3—figure supplement 1*, *Figure 4—figure supplement 1*, *Figure 5—figure supplement 1*, *Figure 3—source data 1*, *Figure 4—source data 1*, *Figure 5—source data 1*). Using the same immune mediator categories defined previously, we first looked at the pro-inflammatory cytokines IL-6, IL-1β, IL-16, CXCL8, TNF, IFN-γ, IL-1α, and LT-α. Notably, two pro-inflammatory cytokines that have been considered master regulators of parturition, IL-6 (*Romero et al., 2006a*; *Robertson et al., 2010*; *Gomez-Lopez et al., 2016*) and IL-1β (*Romero et al., 1989*; *Romero et al., 1992a*), were increased in women who ultimately experienced sPTB compared to those who delivered at term (*Figure 3A and B*). Next, we divided cases of sPTB into sPTL and PPROM, as prior studies have suggested that there are distinct mechanisms of disease for PPROM and sPTL (*Goldenberg et al., 2008*; *Romero et al., 2014a*). We observed that IL-6 and IL-1β were increased in both sPTB subsets compared to women who delivered at term (*Figure 3A and B*). As early and late preterm birth have differing pathophysiology (*Goldenberg et al., 2000*), and adverse outcomes of prematurity are worsened with decreasing gestational age at delivery (*Blencowe et al., 2013*; *Ely and Driscoll, 2020*), we further subdivided cases of sPTL and PPROM into early (<34 weeks) or late (34–36^{+6} weeks) according to gestational age at delivery (*Martin and Osterman, 2018*). While the increase in all and late sPTL cases was consistent across all elevated pro-inflammatory cytokines, an increase in both all PPROM and exclusively early PPROM groups was noted only for IL-6 and IL-1β (*Figure 3A and B*). Likewise, IL-16, a pro-inflammatory mediator (*Chupp et al., 1998*) reported in the female reproductive tract (FRT) (*Athayde et al., 2000*; *Florova et al., 2021*), was also increased in late sPTL and early PPROM (*Figure 3C*). The vaginal concentrations of CXCL8 (high-affinity, HA), TNF, and IFN-γ, immune mediators previously reported in the vaginal fluid with conflicting results (*Imseis et al., 1997*; *Wennerholm et al., 1998*; *Donders et al., 2003*; *Kalinka et al., 2005*; *Chandiramani et al., 2012*; *Kacerovsky et al., 2015*; *Jung et al., 2016*; *Yoo et al., 2017*; *Short et al., 2018*; *Mikołajczyk et al., 2020*; *Short et al., 2021*; *Chan et al., 2022*; *Grewal et al., 2022*), were also increased in women with sPTB compared to term controls (*Figure 3D–F*). Yet, such elevation appeared to be driven by late sPTL cases (*Figure 3D–F*). The vaginal concentrations of CXCL8 (low-affinity, LA), IL-1α, and LT-α were unchanged with sPTB (*Figure 3—figure supplement 1A–C*). Taken together, these results suggest that some sPTB cases are characterized by a pro-inflammatory milieu in the vaginal ecosystem, which may contribute to the pathophysiology of sPTB; yet, the nature of this inflammatory profile is distinct between clinically defined subsets, highlighting the importance of subcategorization of sPTB for obstetrical disease.

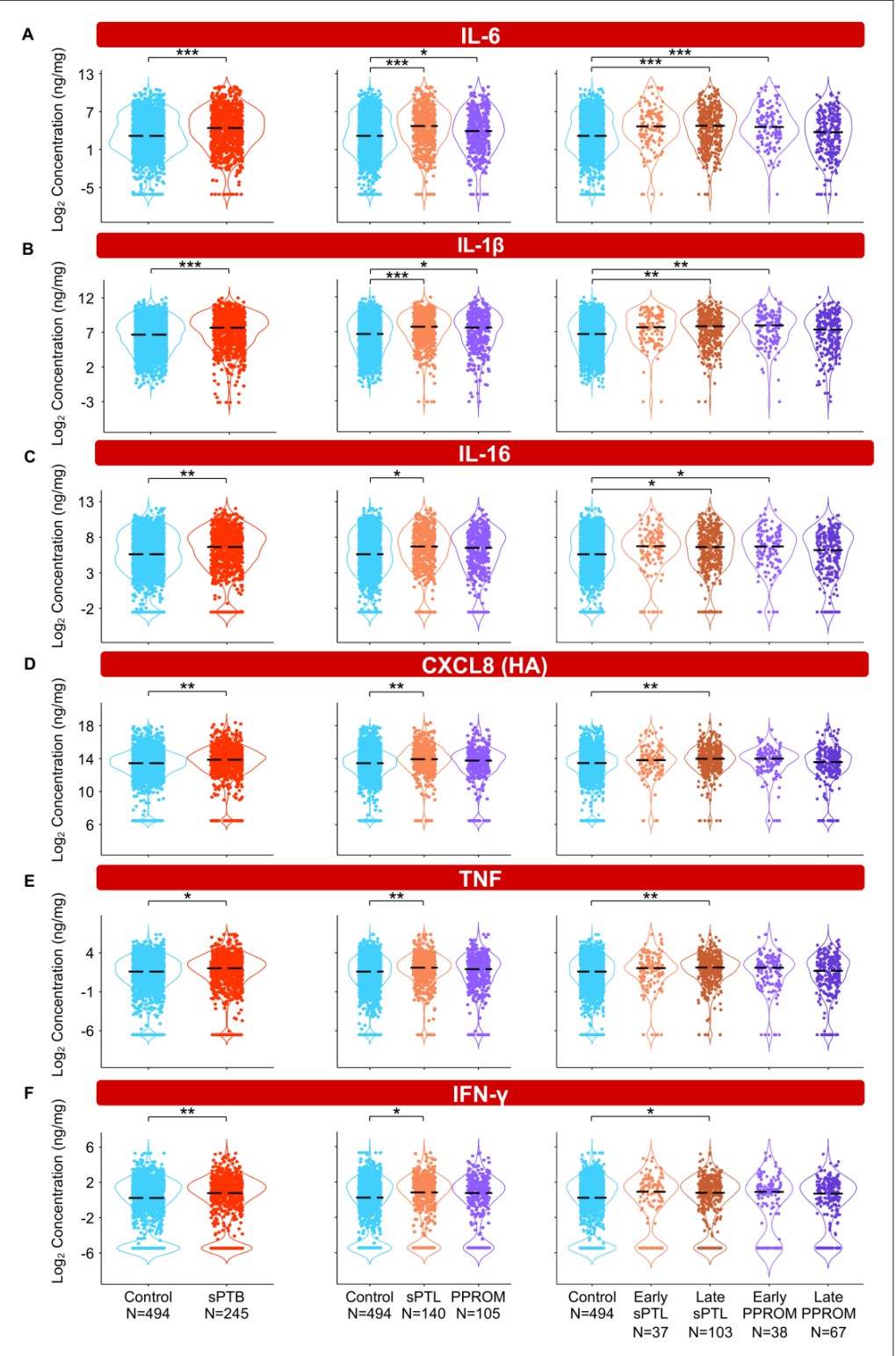

**Figure 3.** The vaginal immunoproteome of women with spontaneous preterm birth (sPTB) displays a pro-inflammatory profile. Linear mixed effects modeling adjusted for gestational age at sampling, body mass index (BMI), parity, and history of preterm birth was used to compare the concentrations of proteins between groups. Violin plots showing the concentrations of (**A**) IL-6, (**B**) IL-1β, (**C**) IL-16, (**D**) CXCL8 (HA), (**E**) TNF, and (**F**) IFN-γ. Violin plots for early spontaneous preterm labor with intact membranes (sPTL) and early preterm prelabor rupture of membranes (PPROM) contain data <34 weeks, all other violin plots contain data <37 weeks. Light blue = controls,

*Figure 3 continued on next page*

*Figure 3 continued*

red = all sPTB, orange = sPTL, purple = PPROM, light orange = early sPTL (gestational age at delivery <34 weeks), dark orange = late sPTL (gestational age at delivery 34–36$^{+6}$ weeks), light purple = early PPROM (gestational age at delivery <34 weeks), dark purple = late PPROM (gestational age at delivery 34–36$^{+6}$ weeks). *p<0.05, **p<0.01, ***p<0.001.

The online version of this article includes the following source data and figure supplement(s) for figure 3:

**Source data 1.** Pro-inflammatory immune mediator concentration differences between term and sPTB cases with and without adjustment for maternal characteristics, between term, sPTL, and PPROM cases, and between term, early sPTL, late sPTL, early PPROM, and late PPROM cases.

**Figure supplement 1.** Specific pro-inflammatory cytokines in the vaginal fluid are unaltered with birth outcome.

## The vaginal immunoproteome of women with sPTB harbors a stereotypical monocyte/macrophage response

The conventional sources of cytokines in an inflammatory response are monocytes and macrophages (*Gordon and Taylor, 2005*; *van de Veerdonk and Netea, 2010*; *Guilliams et al., 2018*). Therefore, we first quantified chemokines responsible for monocyte recruitment such as CCL2, CCL3, and CCL4 (*Shi and Pamer, 2011*; *Robinson et al., 2021*; *Figure 4A*). Interestingly, each of these chemokines followed the same pattern: these mediators were increased in women with sPTB compared to those who delivered at term (*Figure 4B–D*). Specifically, CCL2 and CCL3 were increased in women with late sPTL and those with early PPROM (*Figure 4B and C*), and CCL4 was increased in early and late sPTL as well as in early PPROM (*Figure 4D*).

As a follow-up to the above findings, we also quantified cytokines that are typically produced by tissue-resident macrophages, such as IL-12 (*Trinchieri et al., 2003*; *Vignali and Kuchroo, 2012*) and IL-15 (*Fehniger and Caligiuri, 2001*). We reasoned that the determination of such macrophage-associated cytokines (*Figure 4—figure supplement 1A*) could provide further insight into the role of the local immunological milieu in the pathophysiology of sPTB. Unlike monocyte chemokines, the vaginal concentrations of the macrophage cytokines did not follow a clear pattern. Specifically, the pro-inflammatory IL-12p70 (*Ethuin et al., 2003*; *Taylor et al., 2013*) was increased in both all and late sPTL cases, but not in other subsets of sPTB (*Figure 4—figure supplement 1B*). By contrast, the anti-inflammatory IL-12/IL-23p40 (*Mattner et al., 1993*; *Kato et al., 1996*) was increased only in early PPROM among the sPTB subsets (*Figure 4—figure supplement 1C*). Yet, the vaginal concentrations of IL-15 were increased in late sPTL as well as in early PPROM (*Figure 4—figure supplement 1D*).

These data suggest that part of the pathophysiology of preterm birth may result in an increased recruitment of monocytes into the lower FRT due to elevated chemokines, a finding observed peripartum in mice (*Timmons et al., 2009*) and in humans (*Osman, 2003*). Similarly, an increase in pro-inflammatory cytokines in the vaginal milieu occurs, possibly contributing to monocyte infiltration and/or the activation of resident macrophages (*Benjelloun et al., 2020*), as indicated by an increase in macrophage-associated cytokines. Together with prior studies implicating activated macrophages in the pro-inflammatory milieu accompanying preterm labor and birth (*Xu et al., 2016*; *Gomez-Lopez et al., 2021*), our study further incriminates such an innate immune response in the pathophysiology of a subset of preterm births.

## The vaginal immunoproteome of women with sPTB is enriched for T-cell mediators

Important components of mucosal immunity in the vagina are T cells (*Benjelloun et al., 2020*) and T-cell-associated mediators, such as IL-2 (*Short et al., 2018*; *Gandhi et al., 2020*; *Florova et al., 2021*), IL-4 (*Short et al., 2018*; *Florova et al., 2021*; *Kumar et al., 2021*), IL-17A (*Florova et al., 2021*), IL-10 (*Ashford et al., 2018*; *Short et al., 2018*; *Florova et al., 2021*), IL-13 (*Short et al., 2018*; *Florova et al., 2021*), IL-5 (*Florova et al., 2021*), CXCL10 (*Florova et al., 2021*), and CCL17 (*Florova et al., 2021*), the concentrations of which were determined in the current study (*Figure 5A*). The partial determination of such cytokines/chemokines has been previously reported in the vaginal fluid (*Short et al., 2018*; *Ashford et al., 2018*; *Gandhi et al., 2020*; *Kumar et al., 2021*); however, their combined evaluation across gestation had not been undertaken. The vaginal concentrations of IL-2 and IL-10 were increased in women with sPTB; specifically, in those with late sPTL, but not in

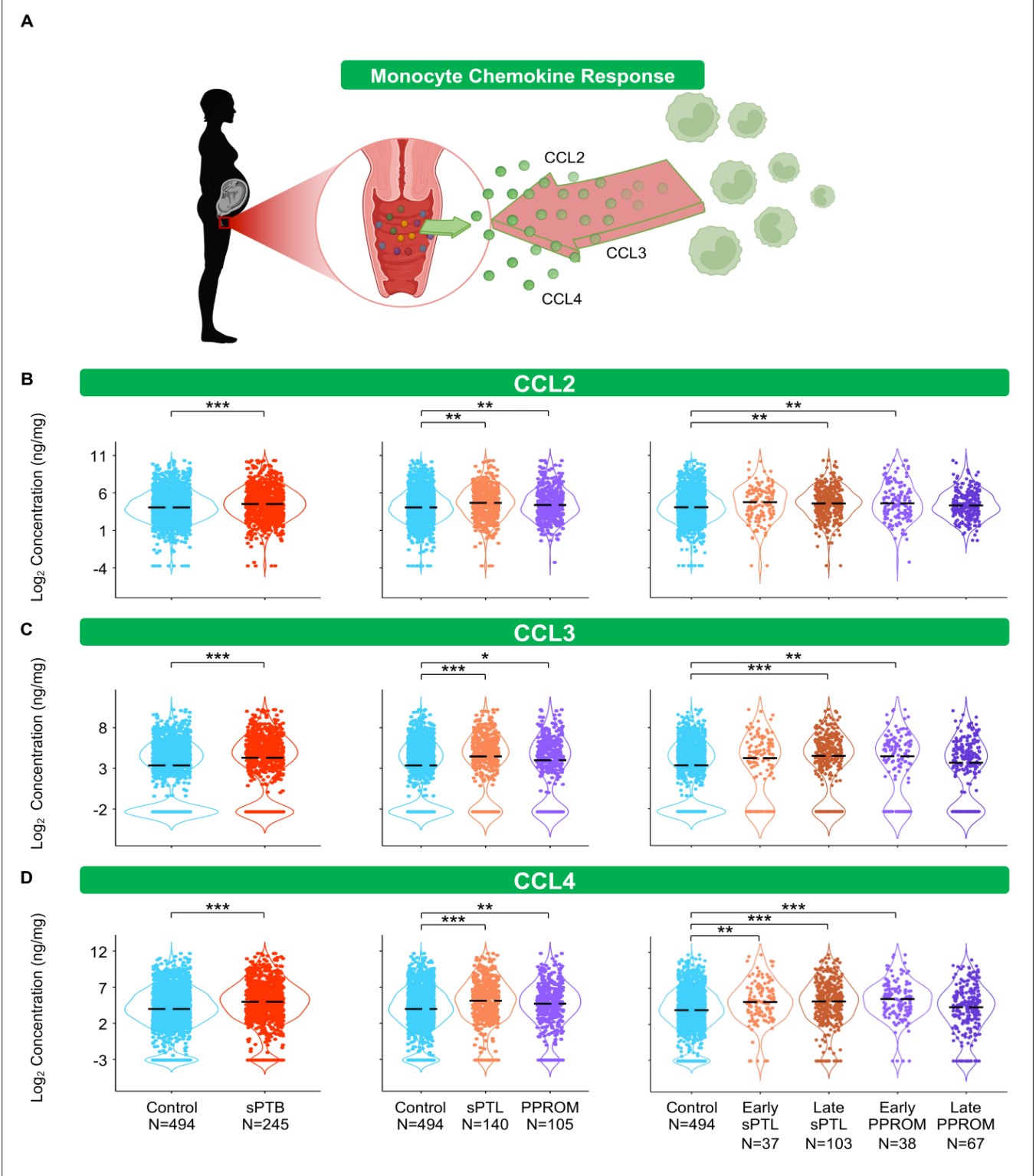

**Figure 4.** The vaginal immunoproteome of women with spontaneous preterm birth (sPTB) harbors a stereotypical monocyte response. (**A**) Linear mixed effects modeling adjusted for gestational age at sampling, body mass index (BMI), parity, and history of preterm birth was used to compare the concentrations of proteins between groups. Violin plots showing the concentrations of (**B**) CCL2, (**C**) CCL3, and (**D**) CCL4. Violin plots for early spontaneous preterm labor with intact membranes (sPTL) and early preterm prelabor rupture of membranes (PPROM) contain data <34 weeks, all other violin plots contain data <37 weeks. Light blue = controls, red = all sPTB, orange = sPTL, purple = PPROM, light orange = early sPTL (gestational

*Figure 4 continued on next page*

*Figure 4 continued*

age at delivery <34 weeks), dark orange = late sPTL (gestational age at delivery 34–36⁺⁶ weeks), light purple = early PPROM (gestational age at delivery <34 weeks), dark purple = late PPROM (gestational age at delivery 34–36⁺⁶ weeks). *p<0.05, **p<0.01, ***p<0.001.

The online version of this article includes the following source data and figure supplement(s) for figure 4:

**Source data 1.** Monocyte chemokine and macrophage cytokine concentration differences between term and sPTB cases with and without adjustment for maternal characteristics between term, sPTL, and PPROM cases as well as between term, early sPTL, late sPTL, early PPROM, and late PPROM cases.

**Figure supplement 1.** The vaginal immunoproteome of women with spontaneous preterm birth (sPTB) includes altered macrophage cytokines.

women with PPROM (*Figure 5B and C*). In addition, the vaginal concentrations of IL-4 and IL-17A were increased in specific subsets of sPTB (late sPTL and early PPROM) (*Figure 5D and E*). However, the concentrations of IL-5, IL-13, CXCL10, and CCL17 did not differ between sPTB cases and controls (*Figure 5—figure supplement 1A–D*). These results suggest that certain T-cell-associated mediators are implicated in the pathogenesis of early PPROM and late sPTL. The finding of products from activated T cells expands on previous research implicating T-cell infiltration and increased markers of inflammation in the amniotic fluid of women experiencing PPROM before 34 weeks of gestation (*Galaz et al., 2020*). Indeed, maternal anti-fetal rejection (infiltration of lymphocytes, including T cells, in the placental tissues; *Kim et al., 2015b*) has been suggested as a mechanism of disease for late sPTB (*Kim et al., 2010*), which is consistent with the increase in T-cell-associated cytokines and chemokines observed in this study.

## The vaginal immunoproteome of women with sPTB exhibits altered antimicrobial proteins and growth factors

Defensins are key players in vaginal host defense (*Cole, 2006*; *Shimoya et al., 2006*; *Cole and Cole, 2008*; *Elovitz et al., 2019*; *Florova et al., 2021*). Indeed, it was previously reported that reduced vaginal concentrations of β-defensin-2 were associated with sPTB (*Elovitz et al., 2019*). Therefore, we next determined the concentrations of two canonical vaginal defensins: SLPI (*Thompson and Ohlsson, 1986*) and β-defensin-2 (*Harder et al., 1997*; *Ganz, 2003*; *Figure 6A*). Notably, the vaginal concentrations of both SLPI and β-defensin-2 were decreased in early PPROM (*Figure 6B and C*). These results show that a specific subset of preterm births, early PPROM, is associated with reduced concentrations of antimicrobial proteins, indicating that a dampened or impaired antimicrobial response may be associated with its pathophysiology.

Besides cytokines, chemokines, and defensins, we also determined the vaginal concentrations of growth factors that are associated with immunobiological functions; namely, GM-CSF (*Hamilton, 2008*; *Becher et al., 2016*), VEGF (*Ferrara et al., 2003*; *Apte et al., 2019*), and IL-7 (*Henney, 1989*; *Barata et al., 2019*; *Figure 6A*). While the vaginal concentrations of GM-CSF were only increased in early PPROM (*Figure 6D*), VEGF was decreased in women with early and late sPTL and early PPROM (*Figure 6E*). The vaginal concentrations of IL-7 did not vary between sPTB cases and controls (*Figure 5—figure supplement 1E*). The differing patterns of association with preterm birth observed in the measured growth factors suggest that, rather than being concomitantly increased or decreased as part of a broad shift in the local inflammatory milieu, each carries out signaling functions that may play a distinct role in the context of different sPTB subsets.

A sub-analysis was performed to evaluate the degree to which adjustment for maternal characteristics affected the immune mediators found significantly different with birth outcome in our model. We compared the results of the adjusted model to a model constructed after adjusting only for gestational age but not for maternal characteristics (*Figure 6—source data 1*). Notably, the lack of adjustment for maternal characteristics confirmed the significance of the immune mediators found in the adjusted model as well as four additional immune mediators, suggesting that adjustment for maternal characteristics provides a more stringent model. In addition, a correlation analysis was also performed between the changes in concentrations of vaginal analytes in sPTL vs. controls and PPROM vs. controls (*Figure 6—figure supplement 1*). Overall, the majority of analytes displayed consistent change between the two sPTB subsets, as demonstrated by positive correlation of fold changes ($r = 0.87$, p<0.0001). VEGF and β-defensin-2 showed the largest decrease in sPTL and PPROM compared

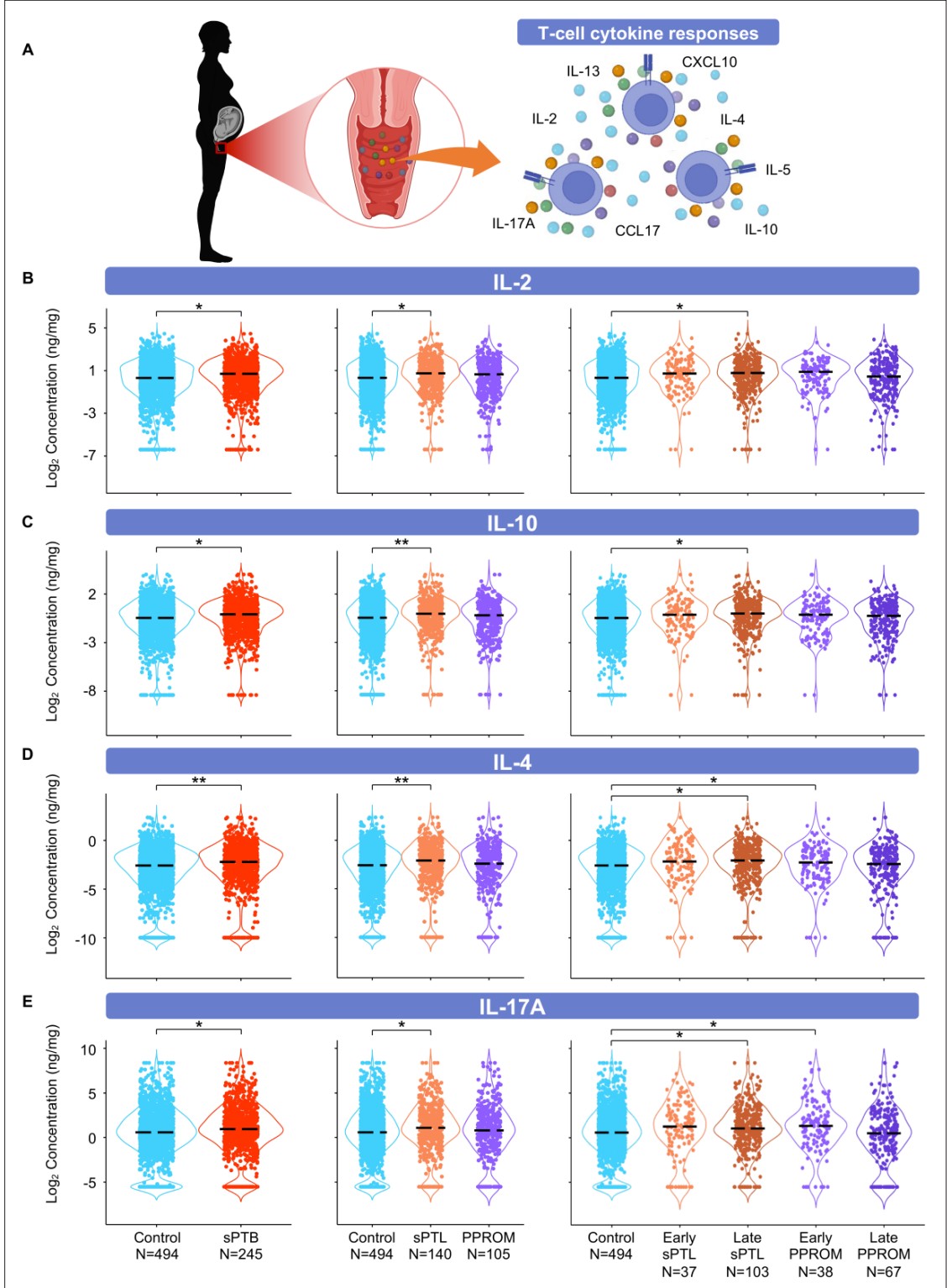

**Figure 5.** The vaginal immunoproteome of women with spontaneous preterm birth (sPTB) is enriched for T-cell mediators. (**A**) Linear mixed effects modeling adjusted for gestational age at sampling, body mass index (BMI), parity, and history of preterm birth was used to compare the concentrations of proteins between groups. Violin plots showing the concentrations of (**B**) IL-2, (**C**) IL-10, (**D**) IL-4, and (**E**) IL-17A. Violin plots for early spontaneous preterm labor with intact membranes (sPTL) and early preterm prelabor rupture of membranes (PPROM) contain data <34 weeks, all other violin plots contain data <37 weeks. Light blue = controls, red = all sPTB, orange = sPTL, purple = PPROM, light orange = early sPTL (gestational

*Figure 5 continued on next page*

The online version of this article includes the following source data and figure supplement(s) for figure 5:

**Source data 1.** T-cell cytokine concentration differences between term and sPTB cases with and without adjustment for maternal characteristics, between term, sPTL, and PPROM cases, and between term, early sPTL, late sPTL, early PPROM, and late PPROM cases.

**Figure supplement 1.** T-cell cytokines/chemokines and the growth factor IL-7 in the vaginal fluid are unaltered with birth outcome.

to controls, whereas CCL3, CCL4, and IL-6 displayed the largest increase. Yet, specific immune mediators (CXCL10, SLPI, and LT-α) displayed differing directions of change between sPTL and PPROM. This finding indicates that sPTB cases share a common inflammatory profile; yet, subtle differences may allow the distinction of its subsets.

## The trajectory of vaginal mediators across gestation differs with birth outcome

Vaginal immune mediators that were shown to be distinct among birth outcome groups in models controlling for gestational age were analyzed again to determine whether the trends of change across gestation varied among the different birth outcomes. Linear mixed effects models with adjustment for all the same covariates as the original analyses, as well as an allowance for an interaction term between sPTB subsets and gestational age at sampling, were constructed to answer this question (*Figure 6—figure supplements 2 and 3*). IL-6, CCL2, CCL3, CCL4, and VEGF were shown to have a significant interaction between the disease outcome for sPTL and PPROM and for gestational age (*Figure 6—figure supplement 2*). Specifically, the trendline for IL-6, CCL2, CCL3, and CCL4 was more positive for the disease groups than in pregnancies resulting in term birth, whereas for VEGF the trendlines for the disease groups were more negative than that for the term birth group. Notably, IL-1β was the only immune mediator that had a significant interaction term (increased trend) for sPTL but not PPROM (*Figure 6—figure supplement 2*), whereas no mediators were significant for PPROM but not sPTL. These data further highlight the shared but subtly different mechanisms of disease underlying sPTL and PPROM.

## The vaginal immunoproteome can be leveraged as a biomarker for early sPTB

Up to this point, our data show that specific components of the vaginal immunoproteome are disrupted in women who ultimately undergo sPTB and that such disruption is distinct among different subsets of preterm birth. Therefore, as a proof of concept, we next evaluated whether the vaginal immunoproteome could serve to generate noninvasive biomarkers for preterm birth subsets with improved predictive value over maternal clinical and demographic information. Random forest modeling of the vaginal immunoproteome was used to establish a predictive tool for sPTB and its subtypes. First, using the last sample taken before 28 weeks of gestation, predictive models were created for sPTB subsets, resulting in delivery after 28 weeks: all sPTB cases (*Figure 7A*), all sPTL cases (*Figure 7B*), sPTL cases delivered before 34 weeks of gestation (sPTL <34 or early sPTL) (*Figure 7C*), all PPROM cases (*Figure 7D*), and PPROM cases delivered before 34 weeks of gestation (PPROM <34 or early PPROM) (*Figure 7E*). When considering the vaginal concentrations of cytokines, chemokines, growth factors, and defensins without including maternal characteristics (black line), the vaginal immunoproteome has the highest potential predictive value for early PPROM <34 (area under the curve [AUC] = 0.787) (*Figure 7E*), with the two most informative immune mediators for the prediction of early PPROM being VEGF and IL-1β. Importantly, limiting inclusion in our model to only the five most significant immunoproteins from a univariate analysis for early PPROM would have resulted in similar performance [AUC = 0.701 (0.6–0.802)]. Nonetheless, the combination of all analytes determined in the vaginal fluid resulted in a model with a lower predictive strength for early sPTL <34 (AUC = 0.637) (*Figure 7C*) compared to early PPROM. For early sPTL, concentrations of IL-1β, IL-16, and IL-13 were most informative. When maternal characteristics (maternal age, BMI, parity, and history of

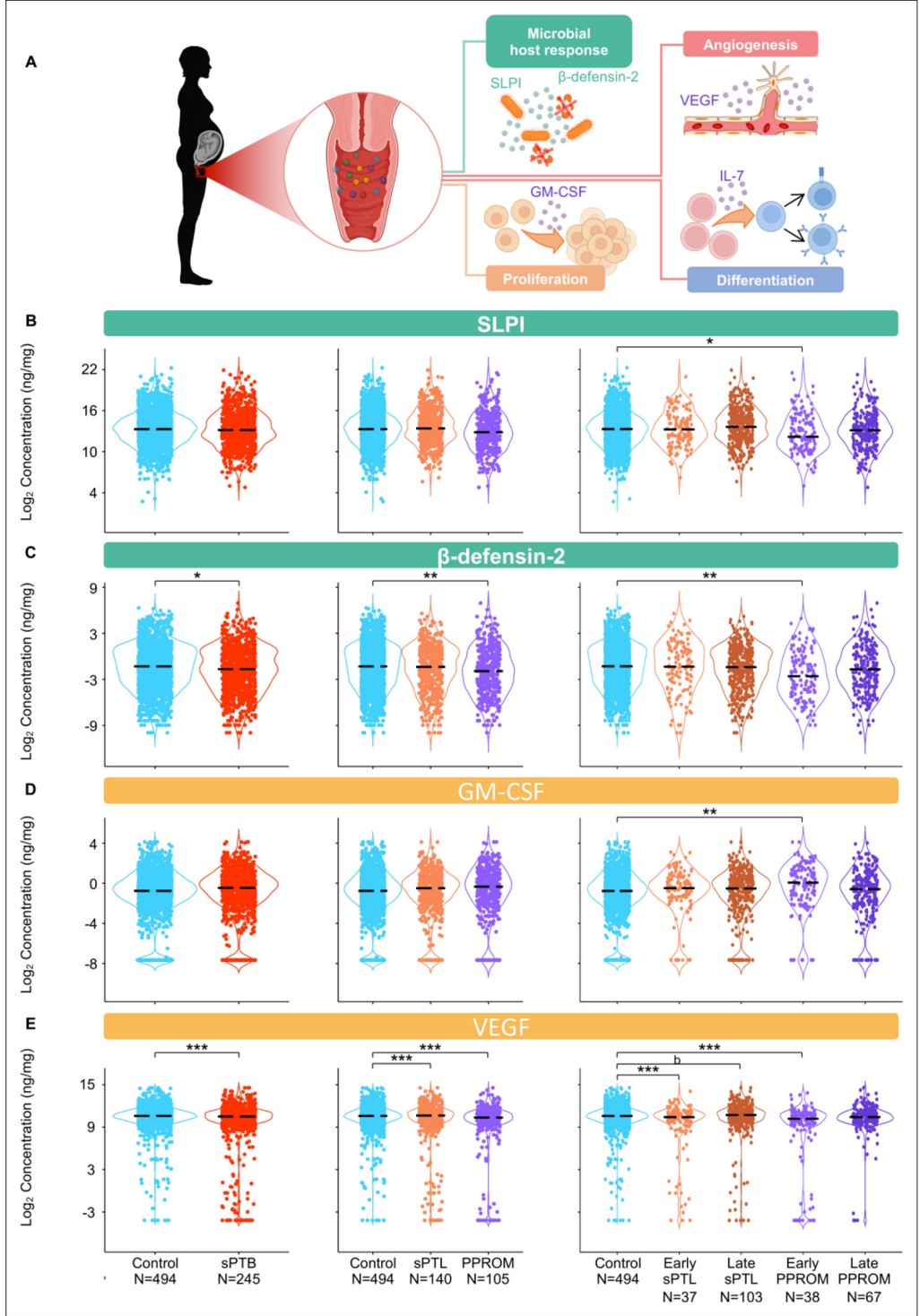

**Figure 6.** The vaginal immunoproteome of women with spontaneous preterm birth (sPTB) exhibits altered antimicrobial proteins and growth factors. (**A**) Linear mixed effects modeling adjusted for gestational age at sampling, body mass index (BMI), parity, and history of preterm birth was used to compare the concentrations of proteins between groups. Immune mediator labels denote category: antimicrobial peptides (AMPs) (teal) and growth factors (orange). Violin plots showing the concentrations of (**B**) SLPI, (**C**) β-defensin-2, (**D**) GM-CSF, and (**E**) VEGF. Violin plots for early spontaneous preterm labor with intact membranes (sPTL) and early preterm prelabor rupture of membranes (PPROM) contain data <34 weeks, all other violin plots contain data <37 weeks. Light blue = controls, red = all sPTB, orange = sPTL, purple = PPROM, light orange = early sPTL (gestational age at delivery <34 weeks), dark orange = late sPTL (gestational age at delivery 34–36$^{+6}$ weeks), light purple = early

*Figure 6 continued on next page*

*Figure 6 continued*

PPROM (gestational age at delivery <34 weeks), dark purple = late PPROM (gestational age at delivery 34–36$^{+6}$ weeks). *p<0.05, **p<0.01, ***p<0.001, $^b$p>0.05 and q<0.1.

The online version of this article includes the following source data and figure supplement(s) for figure 6:

**Source data 1.** Antimicrobial peptide and growth factor concentration differences between term and sPTB cases with and without adjustment for maternal characteristics, between term, sPTL, and PPROM cases, and between term, early sPTL, late sPTL, early PPROM, and late PPROM cases.

**Figure supplement 1.** Changes in the vaginal immunoproteome are correlated between spontaneous preterm labor with intact membranes (sPTL) and preterm prelabor rupture of membranes (PPROM).

**Figure supplement 2.** The trajectory of specific vaginal immune mediators across gestation is distinct with negative birth outcomes compared to gestations resulting in term birth.

**Figure supplement 3.** The trajectory of specific vaginal immune mediators across gestation is unaltered with birth outcome.

---

preterm birth) were included in the models with all vaginal analytes (red line), the prediction accuracy of models predicting sPTB, and the subsets of all sPTL cases and all PPROM cases, was improved (p<0.05 for all) (*Figure 7A, B, and D*). On the other hand, the prediction of early sPTL (*Figure 7C*) and early PPROM (*Figure 7E*) was not improved with the addition of maternal characteristics to the predictive model. Similar conclusions about the significance of the predictions were derived using the area under the precision recall curve (AUPR) instead of the AUC (*Figure 7—source data 1*), and by determining empirical confidence intervals from permutations rather than relying on the DeLong confidence intervals for the AUC displayed in *Figure 7*. To show that the vaginal immunoproteome is capturing variations distinct from maternal characteristics, we compared our models, including the vaginal immunoproteome, to models generated from maternal characteristics alone. Immune mediator data and maternal characteristics resulted in significantly improved prediction of PPROM <34 (DeLong test p<0.023) and led to higher point estimates for all other outcomes when using the last sample before 28 weeks (*Figure 7*).

In an effort to determine whether sPTB can be predicted earlier in the second trimester, a new set of models was developed by using samples collected before 24 weeks of gestation to predict preterm birth after 24 weeks (*Figure 7—figure supplement 1*). Utilizing earlier data also allowed for the prediction of earlier sPTB, specifically for delivery before 30 weeks of gestation. When only considering cytokines, chemokines, growth factors, and defensins in the vaginal fluid (black line), an improved predictive value for sPTL <30 (AUC = 0.743) (*Figure 7—figure supplement 1C*) and a similar value for PPROM <30 (AUC = 0.755) (*Figure 7—figure supplement 1E*) were observed compared to predictive values for sPTL and PPROM <34 weeks. Similar to our models for preterm birth <34 weeks, the inclusion of maternal characteristics (red line) in predictive models generated from the samples collected before 24 weeks of gestation improved the prediction accuracy for sPTB, all sPTL cases, and all PPROM cases (p<0.05 for all) (*Figure 7—figure supplement 1A, B and D*), but not early sPTL or early PPROM (*Figure 7—figure supplement 1C and E*). Likewise, the combination of immunoproteomic data and maternal characteristics resulted in higher point estimates than maternal characteristics alone for all outcomes considered, and significantly increased prediction of PPROM <30 (DeLong test p<0.04), when using the last sample before 24 weeks (*Figure 7—figure supplement 1E*).

Next, we compared the prediction of sPTB by the vaginal immunoproteome against quantitative sonographic cervical length. This analysis was based on a subset of 439 controls and 220 sPTB cases who had one or more cervical length scans at the same time as, or at most within 1 week of, vaginal swab collection. As shown in *Figure 7—figure supplement 2*, the vaginal immunoproteome and cervical length provide comparable predictive value for early PPROM; yet, cervical length alone was a better predictor of sPTB than the vaginal immunoproteome. Notably, for early PPROM, adding the immunoproteome data to cervical length increased the AUC point estimate from AUC = 0.727 (0.597–0.857) to AUC = 0.797 (0.678–0.917) (*Figure 7—figure supplement 2E*), suggesting that the

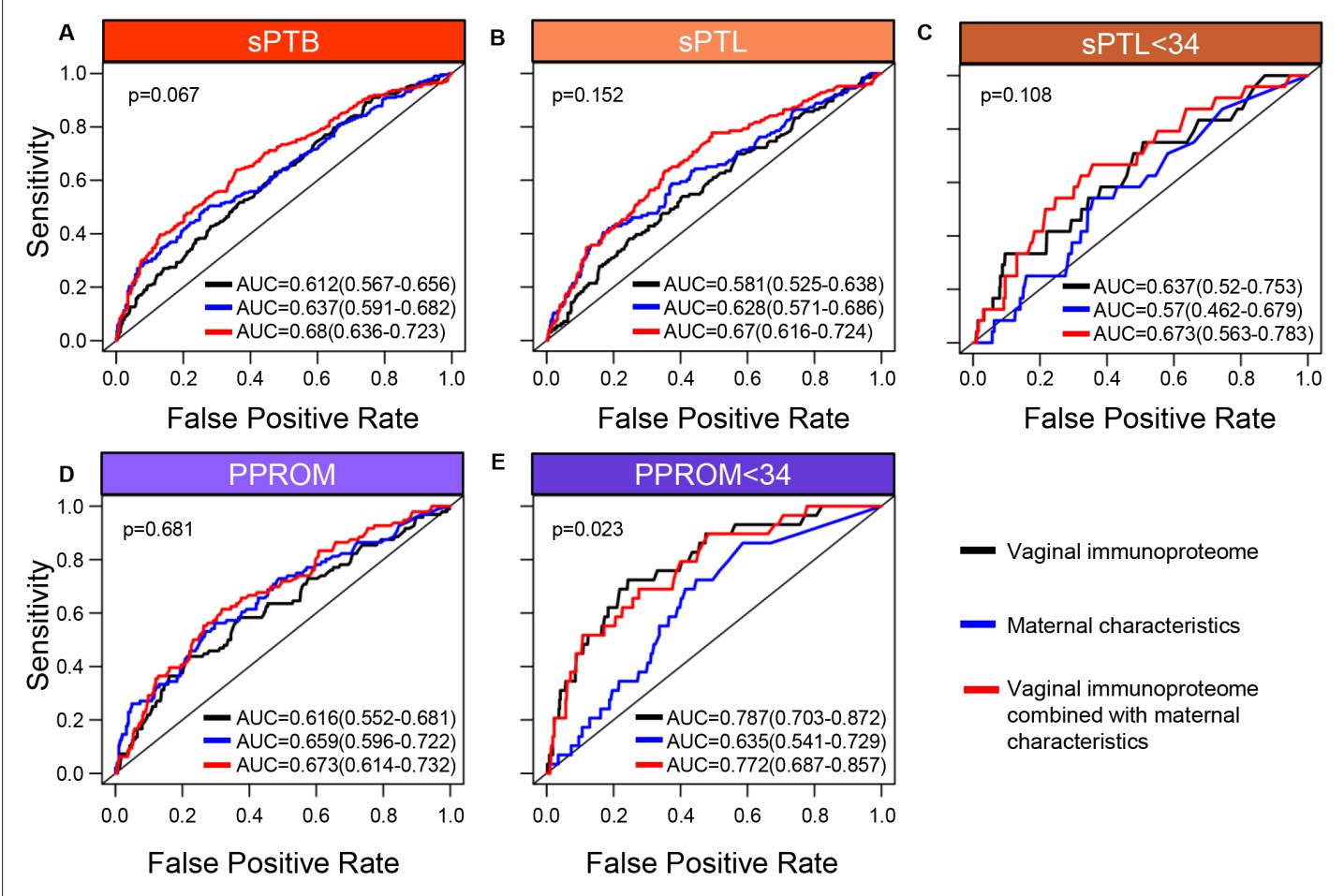

**Figure 7.** The vaginal immunoproteome allows for the prediction of early spontaneous preterm births (sPTB). Random forest modeling including protein concentrations generated from the last swab before 28 weeks of gestation was used to generate models using the combination of all cytokines, chemokines, immune-related growth factors, and antimicrobial molecules in this study (black line), maternal characteristics (blue line), or the combination of all cytokines, chemokines, immune-related growth factors, and antimicrobial molecules in this study together with maternal characteristics (red line) for (**A**) all sPTB, (**B**) all spontaneous preterm labor with intact membranes (sPTL), (**C**) sPTL who delivered before 34 weeks of gestation, (**D**) all preterm prelabor rupture of membranes (PPROM), and (**E**) PPROM who delivered before 34 weeks of gestation. Area under the curve (AUC) values and 95% confidence intervals are given for each curve. P-values correspond to the comparisons between the curves of maternal characteristics alone and the curves of maternal characteristics together with all cytokines, chemokines, immune-related growth factors, and antimicrobial molecules within each study group.

The online version of this article includes the following source data and figure supplement(s) for figure 7:

**Source data 1.** Comparison of AUC vs. AUPR statistics for vaginal immunoproteome predictive models of sPTB and subtypes.

**Figure supplement 1.** The vaginal immunoproteome shows moderate predictive value for early spontaneous preterm labor with intact membranes (sPTL) and preterm prelabor rupture of membranes (PPROM) using swabs collected before 24 weeks.

**Figure supplement 2.** The vaginal immunoproteome provides increased value when combined with the quantitative cervical length measurement model for predicting early preterm prelabor rupture of membranes (PPROM).

incorporation of the vaginal immune response into predictive screening may bolster the strength of these models.

Taken together, these results demonstrate that the vaginal immunoproteome analyzed during the second trimester can generate noninvasive biomarkers with potential predictive value for early preterm birth cases, although the clinical utility of such models needs to be investigated further. Regardless, these data show that the vaginal immunoproteome has improved predictive value over maternal characteristics alone for a subset of early sPTB, highlighting that vaginal immune mediator changes in a subset of preterm birth are attributable to factors other than maternal characteristics.

## Discussion

The vaginal ecosystem includes cellular and soluble components that maintain homeostasis and provide defense against potential pathogens (*Donders et al., 2003*; *Lee et al., 2015*; *Smith and Ravel, 2017*; *Mei et al., 2019*; *Hezelgrave et al., 2020*; *Monin et al., 2020*). While the cellular fraction has been poorly investigated, the soluble components have been well characterized during pregnancy (*Wennerholm et al., 1998*; *Donders et al., 2003*; *Chandiramani et al., 2012*; *Amabebe et al., 2018*; *Ashford et al., 2018*; *Fettweis et al., 2019*; *Witkin et al., 2019*; *Florova et al., 2021*; *Chan et al., 2022*). The soluble fraction includes cytokines, chemokines, antimicrobial peptides/proteins, metabolites, antibodies, and complement components, among others, (*Donders et al., 2003*; *Wei et al., 2010*; *Tribe, 2015*; *Oh et al., 2019*; *Pruski et al., 2021*; *Delgado-Diaz et al., 2022*). It is likely that the vaginal soluble fraction mirrors the three inflammatory phases described at the maternal-fetal interface: a pro-inflammatory profile accompanies the process of implantation, an anti-inflammatory state is maintained throughout the majority of gestation, and an inflammatory milieu is associated with the onset of parturition (*Gomez-Lopez et al., 2010*; *Erlebacher, 2013*). Yet, such a concept has not been established and thus remains to be proven. Herein, we report the largest longitudinal investigation of the vaginal immunoproteome from 8 weeks (i.e., after implantation) to term (collected prior to the onset of parturition), showing that several immune mediators are modulated throughout normal gestation. Specifically, we report that, while multiple innate-derived pro-inflammatory mediators undergo a steady reduction, T cell-associated cytokines and the pro-angiogenic factor VEGF increase throughout gestation. These findings suggest that the vaginal immunoproteome undergoes a modest modulation to help ensure that a homeostatic microenvironment is maintained. Our results are consistent with the prevailing hypothesis that immune homeostasis is required to maintain a successful pregnancy until delivery, and that a breakdown of such homeostasis is implicated as a mechanism of disease for sPTB (*Gomez-Lopez et al., 2020*; *Gomez-Lopez et al., 2021*; *Gomez-Lopez et al., 2022a*). Consistent with this hypothesis, systems biology approaches using proteomics, cytomics, and transcriptomics have conceptualized an immunological clock for pregnancy and have suggested that the early detection of its malfunction may allow for the prediction of sPTB (*Aghaeepour et al., 2017*; *Aghaeepour et al., 2018*; *Gomez-Lopez et al., 2019c*; *Ghaemi et al., 2019*; *Peterson et al., 2020*; *Tarca et al., 2021*; *Stelzer et al., 2021*; *Gomez-Lopez et al., 2022b*). Herein, we built upon this concept by longitudinally exploring the vaginal immunoproteome of women who ultimately underwent sPTB.

Our first set of results showed that multiple pro-inflammatory cytokines, particularly IL-6 and IL-1β, were upregulated in the vaginal fluid of women who ultimately underwent sPTL or PPROM compared to women who delivered at term. Interleukin-6 is a highly pleiotropic cytokine that participates in acute and chronic inflammation, hematopoiesis, and other developmental and physiological processes (*Jones and Jenkins, 2018*). Importantly, IL-6 not only regulates the innate immune response but also participates in the transition to a sustained adaptive immune response, which has made this cytokine and its family attractive targets for immunotherapies (*Jones and Jenkins, 2018*). Moreover, IL-6 regulates the recruitment and activity of leukocytes and may thus be an important upstream regulator of the vaginal chemokine response observed in women with sPTB in the current study. Interleukin-6 also plays a central role in antimicrobial/antiviral immunity (*Jones and Jenkins, 2018*). Indeed, IL-6 is the established biomarker for the diagnosis of acute intra-amniotic inflammation (*Yoon et al., 2001*), the most well-established causal link to preterm labor and birth (*Romero et al., 2006b*; *Romero et al., 2007*; *Goldenberg et al., 2008*; *Romero et al., 2014a*). Consistent with the pro-inflammatory functions of IL-6, IL-1β is also an acute inflammatory cytokine that may even precede the upregulation of IL-6 (*Hazuda et al., 1988*; *Cahill and Rogers, 2008*). Yet, IL-1β is synthesized as a zymogen and therefore requires processing via specific intracellular machinery (i.e., the NLRP3 inflammasome) to be released in its mature and bioactive form (*Kostura et al., 1989*). Notably, we have provided in vivo and in vitro demonstrations that the tissues surrounding the amniotic cavity (i.e., the chorioamniotic membranes), as well as tissues of the upper FRT (e.g., uterine tissues), express the components of the NLRP3 inflammasome (*Plazyo et al., 2016*; *Gomez-Lopez et al., 2017*; *Gomez-Lopez et al., 2018*; *Gomez-Lopez et al., 2019b*; *Gomez-Lopez et al., 2019a*; *Faro et al., 2019*; *Motomura et al., 2020*; *Motomura et al., 2021*; *Motomura et al., 2022*). In fact, in vivo administration of LPS or alarmins, pathogen-associated molecular patterns, and danger-associated molecular patterns that can activate the NLRP3 inflammasome (*Swanson et al., 2019*) results in the processing of active caspase-1 and the

subsequent release of mature IL-1β, leading to preterm labor and birth (*Gomez-Lopez et al., 2019b*; *Faro et al., 2019*; *Motomura et al., 2020*; *Motomura et al., 2021*; *Motomura et al., 2022*). In line with these results, the intra-amniotic infusion of IL-1β triggers the common pathway of parturition and leads to preterm birth in non-human primates (*Gravett et al., 1994*; *Witkin et al., 1994*; *Baggia et al., 1996*; *Vadillo-Ortega et al., 2002*; *Sadowsky et al., 2006*; *Presicce et al., 2015*) and mice (*Yoshimura and Hirsch, 2005*). Moreover, the blockade of either of these pro-inflammatory cytokines through natural inhibitors (e.g., IL-1RA [*Romero and Tartakovsky, 1992b*]) or neutralizing antibodies (e.g., anti-IL-6R [*Wakabayashi et al., 2013*; *Farias-Jofre et al., 2023*]) rescues preterm labor and birth. Thus, given their stereotypical role in acute inflammation, the elevated vaginal concentrations of IL-6 and IL-1β described herein may therefore point to a local inflammatory response that precedes intra-amniotic inflammation that is conventionally associated with sPTB. Whether this local increase represents an aberrant host response to vaginal commensals, the sudden expansion of an opportunistic pathogen, or some other host–microbe interaction requires further investigation. Nonetheless, the above-described evidence further supports IL-6 and IL-1β as master regulators of the onset of preterm parturition, as has been previously demonstrated (*Romero et al., 2006a*; *Robertson et al., 2010*; *Gomez-Lopez et al., 2016*; *Romero et al., 1989*; *Romero et al., 1992a*). It is worth noting that we report distinct patterns of the vaginal immunoproteome between early and late preterm delivery for both sPTB subsets, with the most significant increases observed in women with early PPROM or late sPTL. Thus, these findings provide additional insight into the differing dynamics underlying these two sPTB subsets (*Fortunato et al., 1999*; *Erez et al., 2009*; *Polettini et al., 2014*; *Capece et al., 2014*; *Dutta et al., 2016*; *Dvorakova et al., 2017*; *Menon and Richardson, 2017*). Furthermore, these findings could be partially explained by the genetic predisposition observed in the fetal cord blood of women who undergo PPROM, which displays rare mutations of genes involved in the negative regulation of innate immune activation (*Modi et al., 2017*; *Strauss and Pearson, 2018*); however, additional research is required to investigate the maternal contribution to such a predisposition to undergo PPROM. Regardless of the differing mechanisms that can drive sPTL and PPROM, our data support a shared vaginal immune signature that accompanies inflammation-associated sPTB, potentially reflecting the inflammatory status of the amniotic cavity. This observation is consistent with the concept of a 'common pathway of labor' (*Romero et al., 2014a*) and indicates that it is important to distinguish between changes in vaginal immune mediators that are driven by labor and those that are specific to the disease subset (i.e., sPTL vs. PPROM).

Another finding of the current study was the observed increase in the vaginal concentrations of chemokines implicated in leukocyte recruitment, i.e., CCL2, CCL3, and CCL4, in women who ultimately underwent sPTL or PPROM. The primary receptor for CCL2 is CCR2, which is predominantly expressed by inflammatory monocytes and controls the trafficking of such cells (*Griffith et al., 2014*). CCL2 is rapidly produced by activated tissue or immune cells, and CCL2/CCR2 interactions are considered to be required for inflammatory monocyte migration into peripheral tissues (*Kuziel et al., 1997*). CCL3 and CCL4 can signal through CCR1 and CCR5, which can be expressed by monocytes and T cells (*Griffith et al., 2014*; *Dyer, 2020*). CCR1 and CCR5 have been demonstrated to promote monocyte adhesion and transmigration in an in vitro setting (*Weber et al., 2001*). Indeed, monocytes and macrophages are found in the lower reproductive tract in low numbers (*Wang et al., 2021*), including during pregnancy (*Timmons et al., 2009*), and their abundance is associated with inflammation (*Wang et al., 2021*). In addition, the priming of CD8+ T cells also relies on CCL3/CCL4 signaling through CCR5 (*Castellino et al., 2006*; *Hugues et al., 2007*), thus the increased presence of these chemokines in the vaginal fluid of women who experience sPTB has interesting implications for the pathogenesis of preterm birth. In the context of prior findings, our results suggest that increased chemokine concentrations may contribute to a hostile vaginal milieu through enhanced infiltration of monocytes that could then activate local T cells, a considerable portion of the vaginal immune cell composition (*Lee et al., 2015*). Our findings are in line with a prior report that demonstrated increased cervicovaginal concentrations of CCL2 in women who underwent preterm birth compared to those who delivered at term (*Fettweis et al., 2019*). Moreover, women who ultimately will develop a short cervix, which is a strong predictor of sPTB (*Andersen et al., 1990*; *Iams et al., 1996*; *Heath et al., 1998*; *Hassan et al., 2000*; *Goldenberg et al., 2008*; *Romero et al., 2014a*), displayed a fivefold increase in their vaginal concentrations of CCL2 compared to controls (*Chandiramani et al., 2012*). These data suggest that women destined to undergo sPTL and PPROM share a common signature of monocyte

recruitment prior to disease onset; indeed, such infiltrating cells may represent a source of the acute vaginal cytokine response (i.e., elevated concentrations of IL-6 and IL-1β) that was also observed herein. However, the evaluation of individual mediator kinetics throughout pregnancy would require a substantially greater number of sampling points, making the determination of whether chemokine release precedes the acute inflammatory response nearly impossible. Nonetheless, the role of such mediators in the pathogenesis of each subset of sPTB requires further investigation.

AMPs, which include defensins, cathelicidins, whey acidic proteins, lysozymes, C-type lectins, and S100 proteins, among others, are soluble mediators that participate in the host innate immune response against pathogens in the FRT (*Yarbrough et al., 2015*). Defensins are the largest family of AMPs found in humans, comprising small cationic peptides produced by a variety of immune and nonimmune cells, and are subdivided into α, β, and θ defensins (*Lichtenstein, 1991*; *Yang et al., 2002*; *Maisetta et al., 2003*; *Ganz, 2003*). Specifically, there are four human β-defensins (HBD-1–4) which are primarily expressed by epithelial cells and have been reported in the FRT (*King et al., 2003*; *Yarbrough et al., 2015*), including the vaginal fluid (*Elovitz et al., 2019*; *Florova et al., 2021*) as well as in the gestational tissues (*Espinoza et al., 2003*; *Soto et al., 2007*; *King et al., 2007*; *Kim et al., 2022*). Indeed, the vaginal concentrations of β-defensin-2 were found to be decreased in women who ultimately underwent sPTB compared to those who delivered at term (*Elovitz et al., 2019*). Consistently, in the current study we confirmed that the vaginal concentrations of β-defensin-2 are decreased in women with sPTB, particularly in those with early PPROM. Early PPROM has been previously associated with increased rates of neonatal morbidity (*Goya et al., 2013*; *Yu et al., 2015*; *Pinto et al., 2019*) and mortality (*Goya et al., 2013*; *González-Mesa et al., 2021*; *Yu et al., 2015*), thus women at risk for this subset of sPTB represent a target population for the development of predictive and preventive clinical tools. Such a decrease in β-defensin-2 was mirrored by SLPI, another AMP that has also been reported in the FRT and vaginal fluid (*Yarbrough et al., 2015*; *Florova et al., 2021*). SLPI is a secreted inhibitor that protects host cells against damage from extracellular proteases and is a central player in the constitutive host response in the FRT (*Hein et al., 2002*; *Itaoka et al., 2015*; *Florova et al., 2021*). We previously showed that SLPI is negatively correlated with *Gemella* spp. (*Florova et al., 2021*), a member of the vaginal community state type (CST) IV (*Ravel et al., 2011*) that is linked to increased risk for preterm birth (*DiGiulio et al., 2015*; *Tabatabaei et al., 2019*; *Elovitz et al., 2019*; *Hočevar et al., 2019*; *Chang et al., 2020*; *Odogwu et al., 2021*; *Kumar et al., 2021*; *Dunlop et al., 2021*). Similar to SLPI, reduced vaginal concentrations of β-defensin-2 are associated with vaginal CST IV and sPTB (*Elovitz et al., 2019*). Taken together, these results suggest that the vaginal ecosystem of women who are destined to undergo early PPROM displays a suppressed AMP-driven host response; however, additional research is required to determine whether this decrease reflects an impaired or a dampened response.

A distinct finding is that VEGF, uniquely among the growth factors, was decreased in women who experienced sPTB. VEGF is one member of a family of pro-angiogenic, pro-vasculogenic growth factors with mild immunobiological functions including hematopoiesis (*Gerber and Ferrara, 2003*; *Gerber et al., 2002*) and vascular permeability (*Chen et al., 2012*; *Lal et al., 2001*). Indeed, disruption of VEGF proteins and their receptors is largely implicated in diseases associated with vascular malformations including obstetrical diseases such as preeclampsia (*Tarca et al., 2019*; *Tarca et al., 2022*; *Chaiworapongsa et al., 2004*), a known etiology of iatrogenic preterm birth (*Goldenberg et al., 2008*). Interestingly, while disrupted vascularization is an accepted risk factor of sPTB (*Visser et al., 2021*), the evidence surrounding VEGF's role remains limited. While one study implicated decreased amniotic fluid levels of VEGF in PPROM (*Savasan et al., 2010*), another study found no association between preterm birth and VEGF in the cervico-vaginal space (*Yilmaz et al., 2014*). Yet, more recently, we showed that vaginal concentrations of VEGF were negatively correlated with *Gemella* spp. in women who went on to experience preterm birth (*Florova et al., 2021*). In the context of previous studies, our data suggest that a hostile vaginal milieu may lead to reduced VEGF levels and functions, such as angiogenesis, which could be implicated with early preterm birth regardless of the process of membrane rupture.

A major finding of the current study is that the vaginal immunoproteome sampled prior to 24 weeks of gestation can be utilized to generate noninvasive biomarkers with potential utility for the prediction of sPTB occurring before 30 weeks of gestation. This window of sampling is clinically relevant, given that ultrasound screening is performed between 20 and 24 weeks of gestation and improves

the feasibility of implementing vaginal sampling for cytokine determinations as part of prenatal care. Another advantage of our model is its potential for the prediction of early sPTB, which is associated with more severe neonatal outcomes than those cases occurring after 34 weeks of gestation (*Callaghan et al., 2006*; *Crump et al., 2019*) and may provide the opportunity for personalized patient management to improve adverse perinatal outcomes. It is worth mentioning that the models generated herein showed the best predictive value for early PPROM, which highlights the importance of distinguishing between different subsets of sPTB (i.e., early and late sPTL and PPROM) to improve prediction. Indeed, for pregnancies resulting in early preterm birth, the immunoproteome provides improved predictive strength over previous biomarker models that utilized the vaginal microbiome (*Romero et al., 2022*). Nonetheless, our results as well as those of previous studies suggest that the assessment of the immune response and its interactions with the microbiome (*Fettweis et al., 2019*; *Flaviani et al., 2021*; *Florova et al., 2021*; *Elovitz et al., 2019*) could be considered together with other prenatal screening tools to construct a comprehensive minimally invasive screening tool for preterm birth.

Herein, we also showed that overall maternal characteristics alone have a poor predictability for all subsets of preterm birth; yet, the combination of the vaginal immunoproteome and maternal characteristics improved such predictive capacity. Furthermore, while the predictability of the vaginal immunoproteome alone was not superior to that of the sonographic cervical length, the combination of the vaginal immunoproteome and cervical length improved the predictive value for early PPROM. This is important because there is an imperative need for finding noninvasive biomarkers to predict preterm birth in low-resource areas wherein there is a high risk for this pregnancy complication (*DeFranco et al., 2008*; *Smith et al., 2007*). Although sonographic cervical length remains an excellent predictor for preterm birth, the implementation of cervical length measurement as a universal screening modality for sPTB worldwide has been challenging. Indeed, even developed countries have not been able to enact widespread universal screening of cervical length due to logistic barriers as well as patient refusal to participate in this intrusive procedure (*Temming et al., 2016*; *Pedretti et al., 2017*). From a logistical standpoint, the accurate acquisition and interpretation of cervical length measurements require a highly trained maternal-fetal specialist (*Iams et al., 2013*). As of 2018, only 1570 maternal-fetal medicine specialists were registered in the US workforce (*Wenstrom et al., 2018*) and these specialists are neither distributed equally across the country nor the world, thereby representing a scarce resource. Also, the ultrasound equipment and facilities required to perform cervical length examinations have a prohibitive cost and are unavailable to many women, especially those who are medically underserved, reflecting widespread disparities in medicine (*Haviland et al., 2016*). Furthermore, only 1–2% of pregnant mothers have the clinical finding of a sonographic short cervix in a mid-trimester transvaginal ultrasound screening (*Fonseca et al., 2007*). Among these women, selective administration of vaginal progesterone will reduce the risk of delivery before 37 weeks by half (*Fonseca et al., 2007*). Thus, while this pairing of predictive and preventive tools to sPTB has been modestly effective, the population of pregnant women for whom both cervical length and progesterone administration are beneficial represents a small fraction of the 10% of women who deliver preterm worldwide (*Chawanpaiboon et al., 2019*; *Martin and Hamilton, 2021*). Therefore, investigating biomarkers, including previously suggested immune mediators (e.g. SLPI; *Florova et al., 2021*) and β-defensin-2 (*Elovitz et al., 2019*), that can be easily collected and analyzed in a scalable manner could overcome these shortcomings. As shown here, the addition of the vaginal immunoproteome improved the predictability for specific subsets of preterm birth, specifically early PPROM. An advantage of vaginal sampling is that self-swabbing is now considered an effective way of vaginal fluid collection (*Haguenoer et al., 2014*; *Lunny et al., 2015*); therefore, a screening tool based on vaginal biomarkers may be developed for use in low-resource areas with a scarcity of healthcare providers. Herein, vaginal immune mediators were identified with distinct patterns between women who delivered at term and those who delivered preterm. Collectively, the identified vaginal immune mediators in this study could potentially be incorporated into a comprehensive molecular screening assay capable of supplementing sonographic short cervix screening, and even supplanting it in low-resource areas. Yet, further research is needed to investigate its utility.

A limitation of the current report is that, despite our comprehensive survey of the vaginal immunoproteome, the identification of causal mechanisms driving the changes in specific mediators associated with different subsets of preterm birth is yet to be elucidated. The incorporation of microbiome

and other omics, as well as mechanistic approaches requiring the utilization of animal models, was outside the scope of the research question investigated herein. Yet, we consider that the substantial data generated in this study, together with our predictive models, can serve to spark new investigations focused on targeting key mediators and their relationships with local immune-microbiome interactions in the context of pregnancy complications. Lastly, the population included in this study was primarily composed of African-American women. Additional studies are required to ascertain the generalizability of these results to other populations.

The current study represents the largest longitudinal survey of the vaginal immunoproteome in a population at high risk for sPTB. We report that, throughout uncomplicated gestation, the vaginal immunoproteome harbors a cytokine network that represents a homeostatic profile similar to that observed in other body sites during the second phase of pregnancy. By contrast, the vaginal immunoproteome is skewed toward a pro-inflammatory state in women who ultimately undergo sPTL and PPROM. Such an inflammatory profile includes increased monocyte chemoattractants, cytokines associated with macrophage and T-cell activation, and the consistent reduction of antimicrobial proteins/peptides. Notably, our data show that the vaginal immunoproteome during the second trimester holds predictive value for PPROM before 30 weeks of gestation, indicating that the vaginal immune response can be leveraged as part of a noninvasive approach for the prediction of early sPTB, the leading cause of neonatal morbidity and mortality worldwide.

## Resource availability

### Lead contact

Further information and requests for resources and reagents should be directed to and will be fulfilled by the lead contact, Nardhy Gomez-Lopez (nardhy@wustl.edu).

## Methods

### Experimental model and subject details

#### Human subjects, clinical specimens, and definitions

This was a retrospective longitudinal case–control study evaluating the associations between the vaginal immunoproteome across pregnancy and the incidence of sPTB (i.e., sPTL or PPROM resulting in delivery <37 weeks). Term controls were defined as women who delivered ≥37 weeks of gestation. Patients with twins, fetal malformations, or those with less than three vaginal fluid samples collected throughout pregnancy were excluded. All cases meeting these specific criteria with samples in the biobank were included in the study. Cases were matched, based on ethnicity and maternal age, with term controls at a ratio of 1:2, and samples between these groups were further matched by gestational age at sampling. Samples were collected longitudinally across gestation from each subject under direct visualization from the posterior vaginal fornix using a Dacron swab (Medical Packaging Corp, Camarillo, CA) and a BD ESwab (COPAN ITALIA, Brescia, Italy/Beckton Dickinson, Loveton Circle Sparks, MD). No samples were collected after presentation to the clinic with signs of labor or rupture of membranes to avoid samples being affected by amniotic fluid leakage and/or treatment. Vaginal swabs/supernatants were stored at –80°C until cytokine, chemokine, or other biomarker concentrations were assessed. The study was conducted at the Perinatology Research Branch, an intramural program of the *Eunice Kennedy Shriver* National Institute of Child Health and Human Development, National Institutes of Health, U.S. Department of Health and Human Services, Wayne State University (Detroit, MI), and the Detroit Medical Center (Detroit, MI). The collection and use of human materials for research purposes were approved by the Institutional Review Boards of the National Institute of Child Health and Human Development and Wayne State University (#110605MP2F(RCR)). All participating women provided written informed consent prior to sample collection.

Gestational age was determined by the last menstrual period and confirmed by ultrasound examination, or by ultrasound examination alone when the sonographic determination of gestational age was inconsistent with menstrual dating. sPTL was defined as the presence of regular uterine contractions with a frequency of at least two every 10 min and cervical changes between 20 and 36$^{+6}$ weeks of gestation with intact membranes (as determined by a lack of amniotic fluid pooling in the vagina, negative ferning, or a negative nitrazine test). PPROM was defined as amniorrhexis confirmed by vaginal pooling, ferning, or a positive nitrazine test prior to the onset of labor before 37 weeks of

gestation (*Tricomi et al., 1966*; *Friedman and McElin, 1969*; *Bennett et al., 1993*). This classification gave the following sample breakdown: (i) women who delivered at term (494 patients, 1907 samples); (ii) women who underwent sPTL (140 patients, 521 samples); and (iii) women who experienced PPROM (105 patients, 391 samples). All women included in the sPTL and PPROM groups delivered preterm (<37 weeks of gestation). All samples were collected prior to patients presenting with signs of labor or membrane rupture.

## Method details

### Placental histopathological inflammation

Placentas were examined histologically by perinatal pathologists blinded to clinical diagnoses and obstetrical outcomes according to standardized Perinatology Research Branch protocols (*Kim et al., 2015a*). Briefly, 3–9 sections of the placenta were examined, and at least one full-thickness section was taken from the center of the placenta; others were taken randomly from the placental disc. Acute inflammatory lesions of the placenta (maternal placental inflammatory response [defined as invasion of neutrophils into the chorioamniotic membranes] and fetal placental inflammatory response [acute lesions in the umbilical cord and/or chorionic plate, diagnosed by invasion of neutrophils into these tissues]) were diagnosed according to established criteria (*Kim et al., 2015a*; *Redline, 2015*).

### Determination of immune mediator concentrations in vaginal fluid supernatants

All vaginal swabs were processed and were centrifuged for 10 min at $1300 \times g$ and 4°C. Supernatants from Dacron swabs were stored at −80°C until the determination of cytokine/chemokine concentrations. The V-PLEX Pro-Inflammatory Panel 1 (human), Cytokine Panel 1 (human), and the Chemokine Panel 1 (human) immunoassays (Meso Scale Discovery, Rockville, MD) were used to measure the concentrations of IFN-γ, IL-1β, IL-2, IL-4, IL-6, IL-8, IL-10, IL-12p70, IL-13, and TNF (Pro-inflammatory Panel 1); or GM-CSF, IL-1α, IL-5, IL-7, IL-12/IL-23p40, IL-15, IL-16, IL-17A, LT-α, and VEGF (VEGF-A) (Cytokine Panel 1); or CCL11 (Eotaxin), CCL26 (Eotaxin-3), CXCL8 (IL-8 (HA)), CXCL10 (IP-10), CCL2 (MCP-1), CCL13 (MCP-4), CCL22 (MDC), CCL3 (MIP-1α), CCL4 (MIP-1β), and CCL17 (TARC) (Chemokine Panel 1) in the cervicovaginal fluid according to the manufacturer's instructions. A MESO Quick-Plex SQ 120 (Meso Scale Discovery) was used to read the plates and analyte concentrations were calculated with the Discovery Workbench 4.0 (Meso Scale Discovery). The sensitivities of the assays were 0.368 pg/mL (IFN-γ), 0.152 pg/mL (IL-1β), 0.219 pg/mL (IL-2), 0.028 pg/mL (IL-4), 0.113 pg/mL (IL-6), 0.101 pg/mL (IL-8), 0.054 pg/mL (IL-10), 0.084 pg/mL (IL-12p70), 0.119 pg/mL (IL-13), 0.108 pg/mL (TNF), 0.200 pg/mL (GM-CSF), 0.089 pg/mL (IL-1α), 0.162 pg/mL (IL-5), 0.124 pg/mL (IL-7), 0.408 pg/mL (IL-12/IL-23p40), 0.111 pg/mL (IL-15), 0.478 pg/mL (IL-16), 0.381 pg/mL (IL-17A), 0.095 pg/mL (LT-α), 0.264 pg/mL (VEGF), 0.480 pg/mL (CCL11/Eotaxin), 1.099 pg/mL (CCL26/Eotaxin-3), 22.575 pg/mL (CXCL8/IL-8 (HA)), 0.119 pg/mL (CXCL10/IP-10), 0.072 pg/mL (CCL2/MCP-1), 0.230 pg/mL (CCL13/MCP-4), 1.740 pg/mL (CCL22/MDC), 0.319 pg/mL (CCL3/MIP-1α), 0.226 pg/mL (CCL4/MIP-1β), and 0.375 pg/mL (CCL17/TARC).

BD Eswabs were utilized to determine the vaginal fluid supernatant concentrations of SLPI and β-defensin-2 using the Quantikine ELISA human SLPI immunoassay (R&D Systems, Minneapolis, MN) and the β-defensin-2 ELISA immunoassay (ALPCO, Salem, NH), according to the manufacturer's instructions. A SpectraMax iD5 (Molecular Devices, San Jose, CA) was used to read the plates and analyte concentrations were calculated with the SoftMax Pro 7 (Molecular Devices). The sensitivities of the assays were 20.336 pg/mL (SLPI) and 0.007 ng/mL (β-defensin 2).

Vaginal fluid cytokine, chemokine, and other immune mediator concentrations were adjusted by total protein concentration, which were determined using the Pierce BCA Protein Assay Kit (Thermo Fisher Scientific, Rockford, IL), following the manufacturer's instructions. All immunoassay kits were validated for vaginal swab determinations of the analytes.

## Quantification and statistical analysis

Values below detection limit were imputed with 99% of the smallest detected value. Values larger than two times the 99th percentile were set to two times the 99th percentile. All analyses were performed in R (version 3.6.1; *R Development Core Team, 2010*). Plots were generated with ggplot2 (version 3.3.5) and ggpubr (version 0.4.0.). Changes in immunoprotein abundance ($\log_2$ thereof) with

gestational age at sampling were assessed via linear mixed effects models (*Bates and Machler, 2015*) to account for repeated observations from the same individuals using the lme4 package (version 1.1-26). The trends in scatterplots were estimated using generalized additive models with spline transformations of gestational age at sample collection using the mgcv package (version 1.8-35). Differences between sPTB, sPTL, or PPROM and controls were assessed based on data collected at <37 weeks. Comparisons between early (delivery <34 weeks) sPTL or PPROM and controls were based on data collected before 34 weeks. Violin plots for controls, sPTB, sPTL, PPROM, late sPTL, and late PPROM include data collected at <37 weeks. Violin plots for early sPTL and early PPROM contain data <34 weeks. Differences between cases and controls were assessed using linear mixed effects models with adjustment for gestational age at sampling, maternal age, BMI, parity, and history of preterm birth. p-Values for the significance of the coefficients in the linear mixed effects models were determined via *t*-statistics using Satterthwaite's methods for degrees of freedom, which is the default method in the lmerTest package in R (version 3.1.3). To account for multiple testing, nominal p-values were adjusted using the false discovery rate method to obtain q-values. A 10% cutoff of false discovery rate was used in all analyses to infer significance. Prediction of preterm birth after 24 weeks was based on data collected prior to 24 weeks. Prediction of preterm birth after 28 weeks was based on data collected up to 28 weeks. Random forest models (*Breiman, 2001*) (R package randomForest version 4.6–14) with 1000 trees were trained and evaluated via tenfold cross-validation, and AUC was calculated with DeLong 95% confidence intervals (R package pROC version 1.17.0.1). Models with an AUC 95% confidence interval with a lower bound above 0.5 were considered significant. The importance of individual immune mediators in these analyses was determined using the mean decrease in Gini coefficient.

## Acknowledgements

We thank the physicians, nurses, and research assistants from the Center for Advanced Obstetrical Care and Research, Intrapartum Unit, Perinatology Research Branch Clinical Laboratory, and Perinatology Research Branch Perinatal Translational Science Laboratory for help with collecting and processing samples. We also thank Rona Wang and Gregorio Martinez III for help with carrying out some of the assays, and Derek Miller for his critical feedback on the manuscript. This research was conducted by the Perinatology Research Branch, Division of Obstetrics and Maternal-Fetal Medicine, Division of Intramural Research, *Eunice Kennedy Shriver* National Institute of Child Health and Human Development, National Institutes of Health, US Department of Health and Human Services (NICHD/ NIH/DHHS) under contract HHSN275201300006C. ALT, KRT, and NGL were supported by the Wayne State University Perinatal Initiative in Maternal, Perinatal and Child Health. RR contributed to this work as part of his official duties as an employee of the US federal government. Figures include art created with BioRender.com.

## Additional information

### Funding

| Funder | Grant reference number | Author |
|---|---|---|
| Eunice Kennedy Shriver National Institute of Child Health and Human Development | HHSN275201300006C | Roberto Romero |

The funders had no role in study design, data collection and interpretation, or the decision to submit the work for publication.

### Author contributions

Zachary Shaffer, Data curation, Formal analysis, Investigation, Visualization, Methodology, Writing – original draft, Writing – review and editing; Roberto Romero, Conceptualization, Funding acquisition, Methodology, Writing – review and editing; Adi L Tarca, Conceptualization, Resources, Data curation, Software, Formal analysis, Investigation, Methodology, Writing – review and editing; Jose

Galaz, Investigation, Visualization, Methodology, Writing – original draft, Writing – review and editing; Marcia Arenas-Hernandez, Data curation, Visualization, Methodology, Writing – review and editing; Dereje W Gudicha, Software, Formal analysis, Writing – review and editing; Tinnakorn Chaiwora-pongsa, Eunjung Jung, Manaphat Suksai, Resources, Investigation, Writing – review and editing; Kevin R Theis, Conceptualization, Formal analysis, Investigation, Methodology, Writing – review and editing; Nardhy Gomez-Lopez, Conceptualization, Supervision, Funding acquisition, Investigation, Methodology, Writing – original draft, Project administration, Writing – review and editing

### Author ORCIDs
Adi L Tarca  http://orcid.org/0000-0003-1712-7588
Kevin R Theis  http://orcid.org/0000-0002-8690-7599
Nardhy Gomez-Lopez  https://orcid.org/0000-0002-3406-5262

### Ethics
The collection and use of human materials for research purposes were approved by the Institutional Review Boards of the National Institute of Child Health and Human Development and Wayne State University (#110605MP2F(RCR)). All participating women provided written informed consent prior to sample collection.

### Decision letter and Author response
Decision letter https://doi.org/10.7554/eLife.90943.sa1
Author response https://doi.org/10.7554/eLife.90943.sa2

## Additional files

### Supplementary files
• MDAR checklist

### Data availability
The data generated during this study are available on GitHub (copy archived at *Tarca, 2024*).

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
