## [Editor Report]

This study presents important findings about the vaginal immunoproteome throughout pregnancy in cases of term or preterm birth, showing dynamic differences over time and between birth groups. The findings are solid and convincing in demonstrating the clinical usefulness as informative biomarker material to monitor pregnancy health and risk for preterm birth

---

## [Decision Letter]

**Decision letter after peer review:**

Thank you for submitting your article "The vaginal immunoproteome for the prediction of spontaneous preterm birth: a retrospective longitudinal study" for consideration by *eLife*. Your article has been reviewed by 2 peer reviewers, one of whom is a member of our Board of Reviewing Editors, and the evaluation has been overseen by Wei Yan as the Senior Editor.

Essential revisions (for the authors):

1) Based on the findings reported please elaborate on a potentially shared pathobiology of late PTB and early PPROM (line 497-498).

2) sPTL and PPROM are significantly higher in women with a history of sPTB. Can the authors discuss about the meaning of this association?

*Reviewer #1 (Recommendations for the authors):*

Summary

The authors describe here the dynamics of vaginal immunoproteome during pregnancy by analyzing 31 immune mediators in cervicovaginal samples in women with uncomplicated pregnancies (n=494) and women with sPTB, with a separate analysis for spontaneous preterm labor (sPTL, n=140) and preterm prelabor rupture of membranes (PPROM, n=105). The results show how the vaginal cytokine profile changes as pregnancy progress, contributing to physiological homeostasis in an uncomplicated pregnancy. In contrast, patients undergoing sPTL or PPROM presented deviations from these dynamics towards an inflammatory state through increased monocyte, macrophage, and T-cell activation, and reduction of antimicrobial peptides.

Strengths

This is a comprehensive study of the immunoproteome during pregnancy. It involves a high number of patients, analyzes a wide spectrum of cytokines, chemokines, etc., and contemplates a longitudinal collection of samples covering the three trimesters of pregnancy.

The study is well designed and potential bias has been controlled by matching cases and controls based on ethnicity, maternal age, and gestational age at sampling.

The results are clearly presented, well-interpreted, and discussed in light of the vast literature available about immunity and the onset of labor.

The data presented here could be used for the early prediction of sPTB (before 30 weeks) either alone or in combination with other non-invasive biomarkers or measurements.

Improving early prediction of sPTL and PPROM is an important mission to fight the health and economic burdens of preterm births worldwide.

Weaknesses

The majority of the patients included in this work belong to the African-American community; however, it has been reported that the vaginal microbiota composition differs between races, and these variations may affect the vaginal immune profile in other racial populations. Further studies would be necessary to validate or redefine the sPTB immunoproteome signature in women other than African-American. This limitation has been acknowledged by the authors in the Discussion section.

Some suggestions for the authors that may improve the understanding of the paper:

– Line 151: the authors have analysed 31 immune mediators. How were these 31 molecules selected? Can the authors elaborate on why they chose these 31 and no others?

– Lines 154-159: the 31 molecules are divided into 6 groups, which are the molecules belonging to each of the groups. This information can be found later in the manuscript when results are presented in a subsection, but it would be nice to have this information at the beginning of the section.

– Line 458: change "Interleukin-6" by "IL-6".

– Table 1: sPTL and PPROM are significantly higher in women with a history of sPTB. Can the authors discuss the meaning of this association?

*Reviewer #2 (Recommendations for the authors):*

The authors present a longitudinal analysis of the vaginal immunoproteome (31 absolute concentrations of a hypothesis-driven selection of cytokines, chemokines, growth factors, and anti-microbial peptides) at four timepoints throughout pregnancy in N = 739 women, n = 2,819 samples who delivered a term or preterm birth (PTB) infant following either spontaneous preterm labor (sPTL), or premature preterm rupture of the membranes (PPROM). Aiming to identify proteins associated with preterm birth, the study reveals proinflammatory signatures present in PTB, such as elevated levels of IL1b, IL6, CCL2, CCL3, and decreased levels of VEGF, b-defensin-2, and SLPI, with specific signatures associated with early/late sPTL/PPROM. Longitudinal modeling of the term immunoproteome shows subtle but significant changes in protein levels across term pregnancies, indicative of a precisely-timed regulation of vaginal changes with pregnancy progression. Furthermore, protein measurements before 24, or 28 weeks gestation are informative for classification models to predict PTB vs. term birth, with the strongest predictive performance for PPROM before 34 weeks, and when combined with maternal demographic characteristics, and cervical length measurements.

Overall, this study demonstrates the clinical usefulness of sampling the vaginal microenvironment as accessible, informative material for biomarkers to monitor pregnancy progression and risk for preterm birth. The study can serve as a resource for future investigations and incentivizes to further investigate other pillars of the immune and non-immune environment of the vagina throughout pregnancy, which were outside of the scope of the current study.

This is a neat manuscript.

1. The differences in term vs. preterm, stratified by type and time of PTB, are pointing towards an overlap of late PTB and early PPROM vaginal immunoproteome signature. The authors touch on this finding in the discussion, however, do not further explain it. Please elaborate on a potentially shared pathobiology of late PTB and early PPROM (line 497-498).

2. What do the authors mean by maternal or fetal inflammation (demographics Table 1)? Was there a difference in the immunoproteome based on the presence/absence of an inflammatory state? (see also Discussion line 487-488).

3. The authors report random forest classifiers but do not report which proteins informed the model, and contributed to its performance (despite mentioning that models built using the top univariately different proteins reach a similar performance). Have the authors tried to identify biomarker candidates by using other types of regression models more conducive to feature selection?

4. Comparisons between term and subgroups of PTB cases were assessed based on data collected at <37 or <34 weeks (methods line 786-788). Which term data are shown in the right column for the term group in Figure 3-6, as here, both <37 and <34 weeks comparisons are displayed in the same figure. Furthermore, this information, in addition to in the methods, should be included in text and legends describing the results.

5. Did the authors observe any differences in immunoproteome when stratifying by other demographic categories, e.g., BMI (normal weight, overweight, obese) in healthy term pregnancies?

---

## [Author Response]

Essential revisions (for the authors):1) Based on the findings reported please elaborate on a potentially shared pathobiology of late PTB and early PPROM (line 497-498).

We thank the Reviewer for pointing out the potential for a shared pathobiology underlying late PTB and early PPROM, despite there being differences in the timing during gestation of such changes. It is important to first emphasize that spontaneous preterm birth (sPTB) encompasses both spontaneous preterm labor with intact membranes (sPTL) and preterm prelabor rupture of membranes (PPROM) (1). Both sPTL and PPROM are frequently accompanied by a local (i.e., intra-amniotic) inflammatory response triggered by ascending infection or sterile inflammation (2, 3). Regardless of its nature, such an intra-amniotic inflammatory response comprises elevated concentrations of numerous cytokines (4), including IL-6, IL-1α, and IL-1β, of which IL-6 is used clinically as a biomarker to diagnose intra-amniotic inflammation (5) and IL-1/IL-1β are both causally linked to the onset of labor (6, 7). Importantly, the local inflammation in the amniotic cavity can be reflected in the vaginal fluid (8, 9). Thus, the concept that sPTL and PPROM culminating in sPTB are both accompanied by a vaginal immune response is consistent with the existence of a “common pathway of labor” (10) encompassing those molecular and mechanical processes necessary for labor to occur. We consider that our findings support the existence of a shared vaginal immune response between late sPTL and early PPROM that may also be a component (or reflection) of such a common pathway.

Lines amended in the document: Lines 525-533

Amended text: “Regardless of the differing mechanisms that can drive sPTL and PPROM, our data support a shared vaginal immune signature that accompanies inflammation-associated sPTB, potentially reflecting the inflammatory status of the amniotic cavity. This observation is consistent with the concept of a “common pathway of labor” (10), and indicates that care should be taken to distinguish between changes in vaginal immune mediators that are driven by labor and those that are specific to the disease subset (i.e., sPTL vs. PPROM).”

2) sPTL and PPROM are significantly higher in women with a history of sPTB. Can the authors discuss about the meaning of this association?

We thank the Reviewer for raising this important point. Spontaneous preterm birth (sPTB) encompasses both preterm labor with intact membranes (sPTL) and preterm prelabor rupture of membranes (PPROM) (1). It is well established that a history of preterm birth in prior pregnancy is an important risk factor for the current pregnancy, ranging from 15% to over 50%, depending on the number and gestational age of previous deliveries (1). Although the exact mechanisms for recurrence are not well known, it has been proposed that persistent or recurrent intra-uterine infections may play a role on such increased risk (1, 11). Therefore, the higher prevalence of history of spontaneous preterm birth in the sPTL and PPROM groups compared to the term delivery group is expected.

Lines amended in the document: Lines 153-155

Amended text: “As expected (1), the prevalence of sPTL or PPROM was higher in women with a history of sPTB.”

Reviewer #1 (Recommendations for the authors):SummaryThe authors describe here the dynamics of vaginal immunoproteome during pregnancy by analyzing 31 immune mediators in cervicovaginal samples in women with uncomplicated pregnancies (n=494) and women with sPTB, with a separate analysis for spontaneous preterm labor (sPTL, n=140) and preterm prelabor rupture of membranes (PPROM, n=105). The results show how the vaginal cytokine profile changes as pregnancy progress, contributing to physiological homeostasis in an uncomplicated pregnancy. In contrast, patients undergoing sPTL or PPROM presented deviations from these dynamics towards an inflammatory state through increased monocyte, macrophage, and T-cell activation, and reduction of antimicrobial peptides.StrengthsThis is a comprehensive study of the immunoproteome during pregnancy. It involves a high number of patients, analyzes a wide spectrum of cytokines, chemokines, etc., and contemplates a longitudinal collection of samples covering the three trimesters of pregnancy.The study is well designed and potential bias has been controlled by matching cases and controls based on ethnicity, maternal age, and gestational age at sampling.The results are clearly presented, well-interpreted, and discussed in light of the vast literature available about immunity and the onset of labor.The data presented here could be used for the early prediction of sPTB (before 30 weeks) either alone or in combination with other non-invasive biomarkers or measurements.Improving early prediction of sPTL and PPROM is an important mission to fight the health and economic burdens of preterm births worldwide.WeaknessesThe majority of the patients included in this work belong to the African-American community; however, it has been reported that the vaginal microbiota composition differs between races, and these variations may affect the vaginal immune profile in other racial populations. Further studies would be necessary to validate or redefine the sPTB immunoproteome signature in women other than African-American. This limitation has been acknowledged by the authors in the Discussion section.Some suggestions for the authors that may improve the understanding of the paper:– Line 151: the authors have analysed 31 immune mediators. How were these 31 molecules selected? Can the authors elaborate on why they chose these 31 and no others?

We thank the Reviewer for requesting this clarification. There were both scientific and technical factors to be taken into consideration for the selected immunoassays. The overarching goal was to comprehensively evaluate soluble factors that could be assigned potential functions based on prior literature. To cover some of the best studied cytokines/growth factors (IFN-γ, IL-1β, IL-2, IL-4, IL-6, IL-8, IL-10, IL-12p70, IL-13, TNF, GM-CSF, IL-1α, IL-5, IL-7, IL-12/IL-23p40, IL-15, IL-16, IL-17A, LT-α, and VEGF), we utilized two MSD kits (Pro-Inflammatory Panel 1 and Cytokine Panel 1), which allowed us to measure these mediators with much less sample volume than would be required for conventional ELISA. Similarly, to cover some of the better known chemokines (CCL11, CCL26, CXCL8, CXCL10, CCL2, CCL13, CCL22, CCL3, CCL4, and CCL17), we used the MSD Chemokine Panel 1. For bacterial defensins, multiplex kits are largely unavailable, and thus we needed to be more selective of our targets. The two defensins measured in this study, β-defensin-2 and SLPI, have both been demonstrated to be important vaginal antimicrobial peptides (12, 13, 14, 15, 16). Moreover, prior research had shown that vaginal concentrations of β-defensin-2 were reduced in women with sPTB (16). Thus, we considered that β-defensin-2 and SLPI represented viable targets that would potentially undergo modulation in different sPTB subsets.

In summary, the 31 analytes examined in the current study were selected for their biological relevance, to build upon our prior data, and to allow for efficient use of our samples.

– Lines 154-159: the 31 molecules are divided into 6 groups, which are the molecules belonging to each of the groups. This information can be found later in the manuscript when results are presented in a subsection, but it would be nice to have this information at the beginning of the section.

We thank the reviewer for this suggestion. We have added the immune mediator information to the suggested area.

Lines amended in the document: 165-173

Amended text: “proinflammatory cytokines (IL-6, IL-1β, IL-16, CXCL8, TNF, IFN-γ, IL-1α, and LT-α), chemoattractants of monocytes (herein referred to as monocyte chemokines; CCL2, CCL3, and CCL4), macrophage cytokines (IL-12/IL-23p40, IL-12p70, and IL-15), T-cell cytokines and chemokines (IL-2, IL-4, IL-17A, IL-10, IL-13, IL-5, CXCL10, and CCL17), anti-microbial peptides (β-defensin-2 and SLPI), and growth factors (VEGF and GM-CSF).”

– Line 458: change "Interleukin-6" by "IL-6".

We thank the Reviewer for this suggestion. In this instance, ‘Interleukin-6’ was at the start of the sentence so we wrote it out rather than abbreviate it.

– Table 1: sPTL and PPROM are significantly higher in women with a history of sPTB. Can the authors discuss the meaning of this association?

We thank the Reviewer for this question. An in-depth response to this question can be seen in the essential revisions component of this document above. In brief, previous preterm delivery is an established risk factor for sPTB in subsequent pregnancies. The association of sPTL and PPROM with a history of PTB in this study reflects this increased risk.

Lines amended in the document: Lines 153-155

Amended text: As expected (1), the prevalence of sPTL or PPROM was higher in women with a history of sPTB.

Reviewer #2 (Recommendations for the authors):The authors present a longitudinal analysis of the vaginal immunoproteome (31 absolute concentrations of a hypothesis-driven selection of cytokines, chemokines, growth factors, and anti-microbial peptides) at four timepoints throughout pregnancy in N = 739 women, n = 2,819 samples who delivered a term or preterm birth (PTB) infant following either spontaneous preterm labor (sPTL), or premature preterm rupture of the membranes (PPROM). Aiming to identify proteins associated with preterm birth, the study reveals proinflammatory signatures present in PTB, such as elevated levels of IL1b, IL6, CCL2, CCL3, and decreased levels of VEGF, b-defensin-2, and SLPI, with specific signatures associated with early/late sPTL/PPROM. Longitudinal modeling of the term immunoproteome shows subtle but significant changes in protein levels across term pregnancies, indicative of a precisely-timed regulation of vaginal changes with pregnancy progression. Furthermore, protein measurements before 24, or 28 weeks gestation are informative for classification models to predict PTB vs. term birth, with the strongest predictive performance for PPROM before 34 weeks, and when combined with maternal demographic characteristics, and cervical length measurements.Overall, this study demonstrates the clinical usefulness of sampling the vaginal microenvironment as accessible, informative material for biomarkers to monitor pregnancy progression and risk for preterm birth. The study can serve as a resource for future investigations and incentivizes to further investigate other pillars of the immune and non-immune environment of the vagina throughout pregnancy, which were outside of the scope of the current study.This is a neat manuscript.1. The differences in term vs. preterm, stratified by type and time of PTB, are pointing towards an overlap of late PTB and early PPROM vaginal immunoproteome signature. The authors touch on this finding in the discussion, however, do not further explain it. Please elaborate on a potentially shared pathobiology of late PTB and early PPROM (line 497-498).

We thank the Reviewer for this question. An in-depth response to this question can be found in the essential revisions component of this document above. In brief, sPTL and PPROM can both result in sPTB and are both associated with inflammation in the amniotic cavity. The overlap between the findings for these two pathologies is reflected in their shared inflammatory signatures.

Lines amended in the document: Lines 525-533

Amended text: “Regardless of the differing mechanisms that can drive sPTL and PPROM, our data support a shared vaginal immune signature that accompanies inflammation-associated sPTB, potentially reflecting the inflammatory status of the amniotic cavity. This observation is consistent with the concept of a “common pathway of labor” (10), and indicates that care should be taken to distinguish between changes in vaginal immune mediators that are driven by labor and those that are specific to the disease subset (i.e., sPTL vs. PPROM).”

2. What do the authors mean by maternal or fetal inflammation (demographics Table 1)? Was there a difference in the immunoproteome based on the presence/absence of an inflammatory state? (see also Discussion line 487-488).

We thank the Reviewer for pointing this out. We apologize for our oversight in omitting the definition of maternal or fetal placental inflammatory responses in the methods section of our manuscript. These terms are based on the diagnosis conducted through histopathological examination of the placenta, carried out by perinatal pathologists who were blinded to clinical diagnoses and obstetrical outcomes, in accordance with standardized Perinatology Research Branch protocols (17). Maternal placental inflammatory response refers to acute inflammatory lesions of the placenta, and is defined as the invasion of maternal neutrophils into the chorioamniotic membranes (17). Fetal placental inflammatory response refers to the presence of acute lesions (i.e., neutrophil invasion) in the umbilical cord (funisitis) or chorionic plate (chorionic vasculitis) (17, 18). This information has now been added in the methods section of the revised manuscript.

With respect to whether there is a difference in the vaginal immunoproteome based on the presence/absence of an inflammatory state, when the PTL and PPROM groups were further divided by the presence/absence of maternal or fetal placental inflammatory lesions (instead of by gestational age at delivery) all resulting subgroups demonstrated at least some differences in immune mediators relative to controls (see Appendix Table 1). However, disaggregation by gestational age at delivery better accounted for the vaginal imunoproteomic changes than disaggregation by the presence of maternal or fetal placental inflammatory lesions. Therefore, we have remained reporting it as such.

Lines amended in the document: Lines 765-775

Amended text: “Placental histopathological inflammation – Placentas were examined histologically by perinatal pathologists blinded to clinical diagnoses and obstetrical outcomes according to standardized Perinatology Research Branch protocols (17). Briefly, three to nine sections of the placenta were examined, and at least one full-thickness section was taken from the center of the placenta; others were taken randomly from the placental disc. Acute inflammatory lesions of the placenta [maternal placental inflammatory response (defined as invasion of neutrophils into the chorioamniotic membranes) and fetal placental inflammatory response (acute lesions in the umbilical cord and/or chorionic plate, diagnosed by invasion of neutrophils into these tissues)] were diagnosed according to established criteria (17, 19)

3. The authors report random forest classifiers but do not report which proteins informed the model, and contributed to its performance (despite mentioning that models built using the top univariately different proteins reach a similar performance). Have the authors tried to identify biomarker candidates by using other types of regression models more conducive to feature selection?

We thank the Reviewer for this suggestion. Within the manuscript, we have now identified the principal immune mediators informing key models for early PPROM (VEGF, Il1β) and early sPTL (IL1β, IL16, and IL13). Nevertheless, many immune mediators contributed to these analyses (see Appendix Figure 1 A-E). These immune mediators were identified using the mean decrease in Gini coefficient (i.e., a metric of importance), and these analyses complement the univariate analyses in earlier sections of the Results. A similar conclusion regarding the non-dominance of a single or very few biomarkers was reached using lasso regression with hyperparameter λ being determined by cross-validation. For all outcomes considered, at least 21 of the 28 proteins had non-zero coefficients in the lasso regression models suggesting shared predictive value among proteins.

Lines amended in the document: 374-376; 380-381; 842-844

Amended text: “When considering the vaginal concentrations of cytokines, chemokines, growth factors, and defensins without including maternal characteristics (black line), the vaginal immunoproteome has the highest potential predictive value for early PPROM <34 (AUC = 0.787) (Figure 7E), with the two most informative immune mediators for the prediction of early PPROM being VEGF and IL-1β.”

“Nonetheless, the combination of all analytes determined in the vaginal fluid resulted in a model with a lower predictive strength for early sPTL <34 (AUC = 0.637) (Figure 7C) compared to early PPROM. For early sPTL, concentrations of IL-1β, IL-16, and IL-13 were most informative.”

“The importance of individual immune mediators in these analyses was determined using the mean decrease in Gini coefficient.”

4. Comparisons between term and subgroups of PTB cases were assessed based on data collected at <37 or <34 weeks (methods line 786-788). Which term data are shown in the right column for the term group in Figure 3-6, as here, both <37 and <34 weeks comparisons are displayed in the same figure. Furthermore, this information, in addition to in the methods, should be included in text and legends describing the results.

We thank the Reviewer for this suggestion. To clarify, the left-most column in each panel of Figures 3-6 shows the control group including data <37 weeks. Term group data collected at <34 weeks was used for the comparisons with early sPTL and early PPROM but was not shown independently on the violin plot.

Lines amended in the document: 827-829

Amended text: “Violin plots for controls, sPTB, sPTL, PPROM, late sPTL, and late PPROM include data collected at <37 weeks. Violin plots for early sPTL and early PPROM contain data <34 weeks.”

5. Did the authors observe any differences in immunoproteome when stratifying by other demographic categories, e.g., BMI (normal weight, overweight, obese) in healthy term pregnancies?

We thank the Reviewer for this suggestion. We did assess possible interactions between BMI (categorical variable with three levels as suggested by the Reviewer) and group (sPTBvsControl) on immune mediator levels, while adjusting for gestational age and nulliparity. The q-value was >0.2 for all immune mediators and hence we concluded a lack of significant interactions. We have also assessed the interaction between group and history of preterm birth on immune mediator levels and the results were again negative (q>0.8 for all).

**References**

1. Goldenberg RL, Culhane JF, Iams JD, Romero R. Epidemiology and causes of preterm birth. Lancet. 2008;371(9606):75-84.

2. Romero R, Miranda J, Chaiworapongsa T, Korzeniewski SJ, Chaemsaithong P, Gotsch F, et al. Prevalence and clinical significance of sterile intra-amniotic inflammation in patients with preterm labor and intact membranes. Am J Reprod Immunol. 2014;72(5):458-74.

3. Romero R, Miranda J, Chaemsaithong P, Chaiworapongsa T, Kusanovic JP, Dong Z, et al. Sterile and microbial-associated intra-amniotic inflammation in preterm prelabor rupture of membranes. J Matern Fetal Neonatal Med. 2015;28(12):1394-409.

4. Romero R, Grivel JC, Tarca AL, Chaemsaithong P, Xu Z, Fitzgerald W, et al. Evidence of perturbations of the cytokine network in preterm labor. Am J Obstet Gynecol. 2015;213(6):836.e1-.e18.

5. Yoon BH, Romero R, Moon JB, Shim SS, Kim M, Kim G, et al. Clinical significance of intra-amniotic inflammation in patients with preterm labor and intact membranes. American journal of obstetrics and gynecology. 2001;185(5):1130-6.

6. Romero R, Brody DT, Oyarzun E, Mazor M, Wu YK, Hobbins JC, et al. Infection and labor. III. Interleukin-1: a signal for the onset of parturition. Am J Obstet Gynecol. 1989;160(5 Pt 1):1117-23.

7. Romero R, Mazor M, Brandt F, Sepulveda W, Avila C, Cotton DB, et al. Interleukin-1 α and interleukin-1 β in preterm and term human parturition. American journal of reproductive immunology (New York, NY : 1989). 1992;27(3-4):117-23.

8. Wei SQ, Fraser W, Luo ZC. Inflammatory cytokines and spontaneous preterm birth in asymptomatic women: a systematic review. Obstet Gynecol. 2010;116(2 Pt 1):393-401.

9. Musilova I, Bestvina T, Hudeckova M, Michalec I, Cobo T, Jacobsson B, et al. Vaginal fluid IL-6 concentrations as a point-of-care test is of value in women with preterm PROM. Am J Obstet Gynecol. 2016.

10. Romero R, Dey SK, Fisher SJ. Preterm labor: one syndrome, many causes. Science. 2014;345(6198):760-5.

11. Mazaki-Tovi S, Romero R, Kusanovic JP, Erez O, Pineles BL, Gotsch F, et al. Recurrent preterm birth. Semin Perinatol. 2007;31(3):142-58.

12. Florova V, Romero R, Tarca AL, Galaz J, Motomura K, Ahmad MM, et al. Vaginal host immune-microbiome interactions in a cohort of primarily African-American women who ultimately underwent spontaneous preterm birth or delivered at term. Cytokine. 2021;137:155316.

13. Cole AM. Innate host defense of human vaginal and cervical mucosae. Curr Top Microbiol Immunol. 2006;306:199-230.

14. Shimoya K, Zhang Q, Temma K, Kimura T, Tsujie T, Tsutsui T, et al. Secretory leukocyte protease inhibitor levels in cervicovaginal secretion of elderly women. Maturitas. 2006;54(2):141-8.

15. Cole AM, Cole AL. Antimicrobial polypeptides are key anti-HIV-1 effector molecules of cervicovaginal host defense. Am J Reprod Immunol. 2008;59(1):27-34.

16. Elovitz MA, Gajer P, Riis V, Brown AG, Humphrys MS, Holm JB, et al. Cervicovaginal microbiota and local immune response modulate the risk of spontaneous preterm delivery. Nat Commun. 2019;10(1):1305.

17. Kim CJ, Romero R, Chaemsaithong P, Chaiyasit N, Yoon BH, Kim YM. Acute chorioamnionitis and funisitis: definition, pathologic features, and clinical significance. Am J Obstet Gynecol. 2015;213(4 Suppl):S29-52.

18. Pacora P, Chaiworapongsa T, Maymon E, Kim YM, Gomez R, Yoon BH, et al. Funisitis and chorionic vasculitis: the histological counterpart of the fetal inflammatory response syndrome. J Matern Fetal Neonatal Med. 2002;11(1):18-25.

19. Redline RW. Classification of placental lesions. Am J Obstet Gynecol. 2015;213(4 Suppl):S21-8.